# Prioritizing plant defence over growth through WRKY regulation facilitates infestation by non-target herbivores

**Ran Li[1], Jin Zhang[1], Jiancai Li[1], Guoxin Zhou[1], Qi Wang[1], Wenbo Bian[1], Matthias Erb[2]\*, Yonggen Lou[1]\***

[1]State Key Laboratory of Rice Biology, Institute of Insect Sciences, Zhejiang University, Hangzhou, China; [2]Institute of Plant Sciences, University of Bern, Bern, Switzerland

**Abstract** Plants generally respond to herbivore attack by increasing resistance and decreasing growth. This prioritization is achieved through the regulation of phytohormonal signaling networks. However, it remains unknown how this prioritization affects resistance against non-target herbivores. In this study, we identify *WRKY70* as a specific herbivore-induced, mitogen-activated protein kinase-regulated rice transcription factor that physically interacts with W-box motifs and prioritizes defence over growth by positively regulating jasmonic acid (JA) and negatively regulating gibberellin (GA) biosynthesis upon attack by the chewing herbivore *Chilo suppressalis*. WRKY70-dependent JA biosynthesis is required for proteinase inhibitor activation and resistance against *C. suppressalis*. In contrast, WRKY70 induction increases plant susceptibility against the rice brown planthopper *Nilaparvata lugens*. Experiments with GA-deficient rice lines identify WRKY70-dependent GA signaling as the causal factor in *N. lugens* susceptibility. Our study shows that prioritizing defence over growth leads to a significant resistance trade-off with important implications for the evolution and agricultural exploitation of plant immunity.

\*For correspondence: matthias.
erb@ips.unibe.ch (ME); yglou@
zju.edu.cn (YL)

**Competing interests:** The authors declare that no competing interests exist.

## Introduction

Plants have developed effective defensive systems to minimize herbivore damage. They can specifically perceive attackers and respond to them by activating defence-related signaling pathways, including mitogen-activated protein kinase (MAPK) cascades and hormone signaling, leading to the induction of numerous defence-related genes and defence compounds as well as plant resistance (*Wu and Baldwin, 2010*; *Bonaventure et al., 2011*; *Erb et al., 2012*). Jasmonic acid (JA)-, salicylic acid (SA)-, and ethylene (ET)-mediated signaling play a central role in induced resistance to herbivores (*Lu et al., 2011*; *Qi et al., 2011*).

The induction of defences commonly co-occurs with a reduction of plant growth (*Heinrich et al., 2013*; *Attaran et al., 2014*; *Huot et al., 2014*). Through silencing defence-related genes, a direct negative link between defence and growth was demonstrated (*Zavala and Baldwin, 2006*; *Zhang et al., 2008*; *Meldau et al., 2012*; *Yang et al., 2012*), suggesting that plants actively prioritize defence over growth. Defence prioritization and the associated growth trade-offs are regulated by crosstalk between plant hormones (*Schwachtje et al., 2006*; *Stanton et al., 2013*; *Huot et al., 2014*). DELLA proteins for instance, which typically suppress gibberellin (GA) signaling, can physically interact with the JA pathway repressor JAZ proteins, thereby resulting in mutual suppression (*Hou et al., 2010*; *Yang et al., 2012*). Furthermore, SA has been reported to inhibit the expression of TIR1/ABF F-box genes, thereby leading to stabilization of AUX/IAA repressor proteins and decreasing auxin signaling (*Wang et al., 2007*). Conversely, auxin signaling can reduce SA biosynthesis and thereby render plants more susceptible to pathogens (*Robert–Seilaniantz et al., 2011*). Compared to the impact

**eLife digest** Many different animals feed on plants, including almost half of all known insect species. Some herbivores—like caterpillars for example—feed by chewing. Others, such as aphids and planthoppers, use syringe-like mouthparts to pierce plants and then feed on the fluids within.

To minimize the damage caused by these herbivores, plants activate specific defenses upon attack, including proteins that can inhibit the insect's digestive enzymes. The inhibitors are effective against chewing herbivores but seem to have little or no effect on some insects that feed by the 'pierce-and-suck' method.

Investing in defense requires energy, and so plants attacked by herbivores actively slow their growth to meet this demand. Plants achieve this trade-off by changing the levels of different plant hormones. These hormones can control the expression of thousands of genes and have widespread effects throughout the plant. However, little is known about how prioritizing defense overgrowth in response to an attack by one herbivore affects the plant's ability to defend itself against other herbivores.

Transcription factors are proteins that control which genes inside a cell are active or inactive. Li et al. searched for a transcription factor in rice plants that was specifically triggered in response to an attack by the caterpillars of a moth called the rice striped stem borer. This search identified a protein called WRKY70 as a transcription factor that prioritizes defense overgrowth. WRKY70 achieves this by increasing the levels of a defensive plant hormone (called jasmonic acid) while reducing the levels of a growth hormone (called gibberellin). Further experiments show that the increase in jasmonic acid production is required to activate the enzyme inhibitors and for resistance against these caterpillars.

Li et al. then found that increased WRKY70 activity makes rice plants more susceptible to attack by a second herbivore, a piercing-sucking insect called the rice brown planthopper. Further experiments revealed that this is due to the reduced levels of gibberillin. These findings show that while prioritizing defense overgrowth is effective against some insect herbivores, it comes with a cost as it makes the plants more susceptible to attack by other herbivores. This trade-off has important implications for both the evolution of plant immunity, and efforts to exploit plant immunity to help protect crops from herbivore attack.

of defence-related hormones, little is known about the impact of growth-related hormones on herbivore resistance and potential resistance trade-offs that may emanate from prioritizing defence over growth through hormonal regulation (*Yang et al., 2012*).

Transcription factors (TFs) play a potentially important role in herbivore-induced plant reconfiguration and defence prioritization, as they regulate the expression of responses up and downstream of hormonal signaling pathways and thereby influence early and late signaling (*Reymond et al., 2004*; *Dombrecht et al., 2007*; *Skibbe et al., 2008*; *Kaur et al., 2010*; *Lu et al., 2011*; *Zhou et al., 2011*; *Schweizer et al., 2013*). The best-studied TFs involved in plant–insect interactions are MYCs, WRKYs, MYBs, and ERFs. AtMYC2 in *Arabidopsis thaliana*, for example, was reported to act downstream of JA and to regulate JA-dependent herbivore resistance (*Dombrecht et al., 2007*). Moreover, MYC2, MYC3, and MYC4 were shown to regulate the production of toxic glucosinolates via a direct transcriptional activation of glucosinolate biosynthesis genes (*Schweizer et al., 2013*). In *Nicotiana attenuata,* a R2R3-type MYB TF (*NaMYB8*) was found to modulate the accumulation of phenylpropanoid–polyamine conjugates, which are essential for defence against herbivores (*Kaur et al., 2010*). In rice, an EAR-motif-containing ERF TF (*OsERF3*) functions as an early component upstream of MAPK signaling and modulates JA, SA, ET, and $H_2O_2$ levels as well as plant resistance to rice herbivores (*Lu et al., 2011*). Also, several WRKYs, such as rice OsWRKY89, wheat TaWRKY53, *Arabidopsis* AtWRKY72, and tomato SlWRKY70 and SlWRKY72, have been directly associated with defence against herbivores (*Wang et al., 2007*; *Bhattarai et al., 2010*; *Van Eck et al., 2010*; *Atamian et al., 2012*). NaWRKY3 and NaWRKY6 in *N. attenuata* have been shown to modulate elicited JA and JA-Ile/-Leu levels and thus mediate herbivory-induced defence responses (*Skibbe et al., 2008*). Identifying and manipulating TFs that are involved in defence prioritization would make it possible to assess the biological impact of herbivore-induced growth suppression. Yet, to date, such an approach has not been taken. Consequently, our understanding of the consequences of defence prioritization for plant resistance has remained limited.

To dissect the signaling network that underlies growth defence trade-offs in rice, we identified OsWRKY70, an herbivory-induced Group I-type WRKY TF from rice, and elucidated its role in herbivore-induced defence prioritization. Through the use of in vivo and in vitro protein assays, molecular characterization and the creation of transgenic OsWRKY70 silenced and overexpressing plants combined with insect bioassays and a variety of phytohormone analyses, we evaluate the resistance benefits and trade-offs of defence prioritization against different herbivores and thereby reveal a new cost of defence prioritization.

## Results

### OsWRKY70 is an herbivory-induced, nucleus-localized, auto-regulated W-box transcriptional activator

Using suppressive subtractive hybridization (SSH), we screened rice plants for herbivory-induced TFs. Using this technique, we identified a clone that showed similarity to a WRKY gene. The full-length cDNA of the cloned *OsWRKY*, including an open reading frame (ORF) of 1719 bp, was obtained by reverse transcription PCR (*Figure 1—figure supplement 1*). Blast analysis showed that the sequence was 100% identical to the previously identified *OsWRKY70* (TIGR ID Os05g39720). OsWRKY70 has two WRKY domains and belongs to group I (*Rushton et al., 2010*). Phylogenetic analysis of group I-type WRKYs from different species revealed that OsWRKY70 has two homologs in rice, OsWRKY24 and OsWRKY53, which share 53% and 51% amino acid sequence identity (*Figure 1—figure supplement 2*). Quantitative real-time PCR analysis revealed low constitutive expression of *OsWRKY70*. Mechanical wounding and infestation by the rice striped stem borer (SSB) *Chilo suppressalis* resulted in a rapid increase in transcript levels (*Figure 1A,B*). Infestation by the rice brown planthopper (BPH) *Nilaparvata lugens* only slightly increased the transcription levels of (*Figure 1C*). JA or SA treatment did not induce *OsWRKY70* (*Figure 1D*), suggesting that *OsWRKY70* is an early regulator of plant responses to herbivores.

To clarify the subcellular localization of OsWRKY70, we constructed an *OsWRKY70:GFP* fusion gene, driven by a CaMV 35S promoter, and transiently expressed the construct in *Nicotiana benthamiana* leaves. Fluorescence analysis showed that OsWRKY70 is exclusively localized in the nucleus (*Figure 2—figure supplement 1A*). To determine the DNA-binding activity of OsWRKY70, a His-tagged protein was produced in *Escherichia coli*, and its W-box binding ability was examined by electrophoretic mobility shift assays (EMSAs) as described (*Chujo et al., 2007*). In the presence of the oligonucleotide probe BS65 containing two W-box sequences and a WRKY70 recombinant protein, specific protein-DNA complexes with reduced migration were present in the EMSA assays (*Figure 2—figure supplement 1C*). The DNA-binding specificity was confirmed in a competition experiment using a 250-fold excess of the unlabeled probe BS65 as a competitor, for which no binding complexes were detected. When the W-box in BS65 was mutated from TTGACC to TCCTAC (mBS65), the binding complexes also disappeared. These results indicated that the recombinant OsWRKY70 protein specifically binds to the conserved W-box in the synthesized probe. To investigate whether OsWRKY70 has transcriptional activation activity, we fused the full-length *OsWRKY70* ORF in-frame to the GAL4 DNA-binding domain of the pGBKT7 vector and transformed it into yeast. The yeast transformed with pGBKT7 or pGBKT7-OsWRKY70 was plated on SD medium (−Trp) containing X-α-gal. After 12 hr at 30°C, the pGBKT7-OsWRKY70 transformant yeast colonies turned blue. In contrast, the pGBKT7 empty transformant yeast colonies remained white (*Figure 2—figure supplement 1B*). OsWRKY70 is therefore likely functioning as a transcriptional activator in the yeast system. We found that the promoter region of WRKY70 contains four W-boxes, three reverse W-boxes (AGTCAA at −82 to −77, AGTCAA at −56 to −51, and GGTCAA at −49 to −44), and one forward W-box (TTGACC at −62 to −57), upstream of the transcription start site (*Figure 2—figure supplement 2A*). To investigate if WRKY70 regulates its own expression, we first performed an EMSA assay by using the minimal promoter region of WRKY70 (86 bp upstream of transcription start site) as a probe. WRKY70-His can bind to this fragment, while adding 250-fold unlabeled probe resulted in no WRKY70-DNA complex (*Figure 2—figure supplement 2B*). Using WRKY70 promoter:*GUS* as a reporter and 35S: *WRKY70-GFP*, 35S:*GFP* as effectors expressed transiently in *N. benthamiana*, we found that WRKY70-GFP significantly increased the GUS activities compared to GFP alone, suggesting that WRKY70 can self-activate its transcription (*Figure 2—figure supplement 2C*).

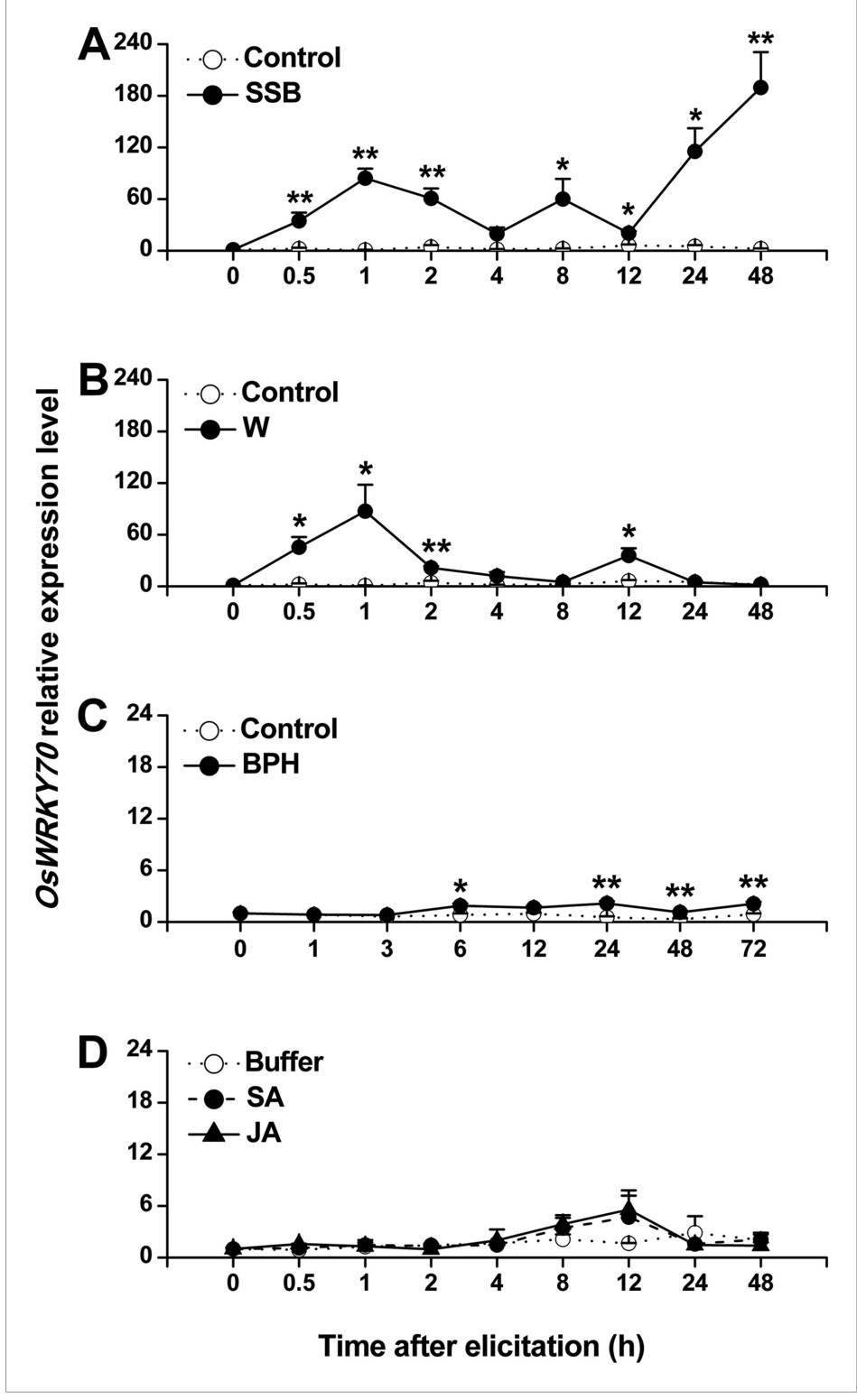

**Figure 1**. Expression of *OsWRKY70* in rice after different treatments. Mean transcript levels (+SE, n = 3–4) of *OsWRKY70* in rice plants that were treated with either rice striped stem borer (SSB) (**A**), mechanically wounded (**B**), rice brown planthopper (BPH) (**C**), jasmonic acid (JA), salicylic acid (SA), or a buffer (50 mM phosphate buffer, pH = 8.0) (Buffer) (**D**). Controls correspond to non-manipulated plants. Transcript levels were analyzed by

*Figure 1. Continued*

QRT-PCR. Asterisks indicate significant differences in transcript levels between treatments and controls (*, $p < 0.05$; **, $p < 0.01$; Student's *t*-test).

The following figure supplements are available for figure 1:

**Figure supplement 1**. Nucleotide and amino acid sequence of OsWRKY70.

**Figure supplement 2**. Phylogenetic relationships of Group I type WRKY genes from different species.

## OsWRKY70 physically interacts with and is regulated by OsMPK3 and OsMPK6

MAPK proteins can specifically recognize the D domain found in some group I-type WRKYs and specifically phosphorylate the Ser residues of Group I SP clusters (*Ishihama et al., 2011*; *Mao et al., 2011*). The D domain is a cluster of basic residues upstream of the LxL motif ([K/R]1–2-x2–6-[L/I]-x-[L/I]) and has been reported in some WRKYs to play an important role in determining the selectivity of interacting MAPKs and phosphorylation patterns (*Ishihama et al., 2011*). OsWRKY70 has four SP clusters in the N-terminal region (*Figure 1—figure supplement 1*) but has no D domains. We hypothesized that the D domain-deficient OsWRKY70 may nevertheless interact with the MAPKs, OsMPK3 and OsMPK6, the homologs of AtMPK3 and AtMPK6 in *Arabidopsis* and WIPK and SIPK in *Nicotiana tabacum*, respectively, all of which are involved in plant defence responses (*Wu et al., 2007*; *Lu et al., 2011*). We used a GST pull-down assay to analyse the interaction between OsWRKY70 and OsMPK3 or OsMPK6 in vitro. OsWRKY70-His was pulled down strongly by GST-MPK3 and mildly by GST-MPK6, suggesting that OsWRKY70 can interact with both OsMPK3 and OsMPK6, with the OsWRKY70–OsMPK3 interaction being more efficient than the OsWRKY70–OsMPK6 interaction in vitro (*Figure 2B*). In vivo, we used a bimolecular fluorescence complementation (BiFC) assay to confirm the interaction. Fluorescence was observed when nYFP-OsWRKY70 was co-injected with OsMPK3-cYFP or OsMPK6-cYFP, and the signals were in the nuclear compartment according to 4,6-diamidino-2-phenylindole (DAPI) staining. No fluorescence was observed when nYFP-OsWRKY70 was co-expressed with unfused cYFP (*Figure 2C*). Taken together, these results strongly suggest that OsMPK3 and OsMPK6 physically interact with OsWRKY70. In a next step, we investigated if OsWRKY70 is phosphorylated by OsMPK3 or OsMPK6. We used OsMKK4$^{DD}$, a constitutively active form of OsMKK4, to activate recombinant OsMPK3 and OsMPK6 and then exposed them to OsWRKY70. The result showed that OsWRKY70 can be phosphorylated by both OsMPK3 and OsMPK6 (*Figure 2D*). Moreover, an EMSA assay revealed that phosphorylation did not alter the W-box-binding activity of OsWRKY70 (*Figure 2E*). We also investigated if phosphorylation enhances the transactivation activity of OsWRKY70 using *N. benthamiana* as a transient expression system (*Li et al., 2014*). As the constitutive expression of OsMKK4$^{DD}$ induces HR-like cell death in tobacco plants, we used an estradiol-inducible system of OsMKK4$^{DD}$ (*Ishihama et al., 2011*). Given that OsWRKY70 can auto-activate its promoter (*Figure 2—figure supplement 2C*), we used WRKY70 promoter:*GUS* as a reporter and 35S:*WRKY70-GFP*, 35S:*OsMPK3-YFP*, 35S:*OsMPK6-YFP* as effectors (*Figure 2F*). GUS activity was higher in OsMKK4$^{DD}$-MPK3-OsWRKY70 or OsMKK4$^{DD}$-MPK6-OsWRKY70 co-expressed leaves than leaves expressing OsWRKY70 alone (*Figure 2E*). These results show that phosphorylation of OsWRKY70 can increase transactivation activity, but not W-box binding activity, of OsWRKY70.

To determine whether the two MAPKs regulate OsWRKY70, we measured transcript levels of *OsWRKY70* in MAPK-silenced rice plants and *vice versa*. *OsWRKY70* transcript levels were significantly decreased in *OsMPK3* (*Wang et al., 2013*) and *OsMPK6* silenced lines (*Figure 3A*) after infestation with SSB for 1 hr (*Figure 2A*). Another group I-type WRKY TF, OsWRKY24, which has both SP clusters and a D domain, was down-regulated in *OsMPK6* silenced plants, but not in *OsMPK3* silenced lines (*Figure 3B*). In *WRKY70* silenced lines (see below) on the other hand, *OsMPK3* and *OsMPK6* transcripts were the same as in wild-type (WT) plants (*Figure 3C,D*). These results show that the transcript levels of *OsWRKY70* is regulated by *OsMPK3* and *OsMPK6*, but not *vice versa*.

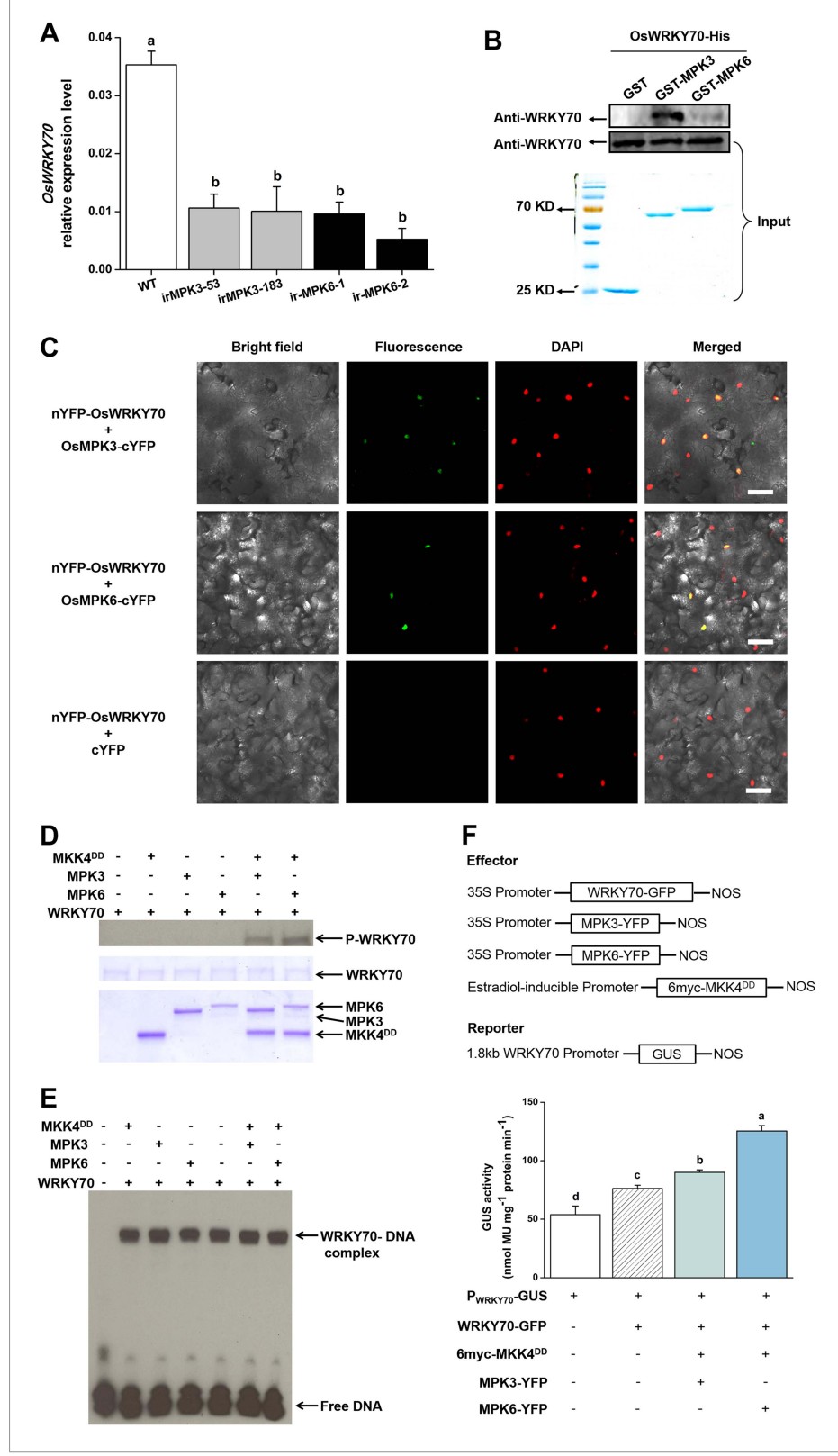

**Figure 2**. Interactions between OsWRKY70 and OsMPK3/6. (**A**) Mean transcript levels (+SE, n = 5) of *OsWRKY70* in transgenic lines with silencing of OsMPK3 (irMPK3 lines, irMPK3-53, and irMPK3-183) or OsMPK6 (irMPK6 lines, irMPK6-1, and irMPK6-2) after infested by SSB for 1 hr. (**B**) In vitro interaction assays between OsWRKY70 and OsMPK3 or OsMPK6. GST, GST-MPK3, and GST-MPK6 purified proteins were incubated with WRKY70-His as

*Figure 2. continued on next page*

*Figure 2. Continued*

indicated. WRKY70-His input and pulled-down fractions were analyzed by immunoblotting using anti-WRKY70 antibody (top). Input proteins were monitored by Coomassie blue staining (bottom). This experiment was repeated 3 times with similar results. (**C**) In vivo bimolecular fluorescence complementation interaction assays between OsWRKY70 and OsMPK3 or OsMPK6. Fluorescence was observed from complementation of the N-terminal part of the YFP fused with OsWRKY70 (nYFP-OsWRKY70) with OsMPK3 or OsMPK6 fused with the C-terminal part of the YFP (OsMPK3-cYFP or OsMPK6-cYFP) and co-localized with DAPI stains in the nuclear compartment of tobacco leaf cells. No fluorescence was observed when nYFP-OsWRKY70 was co-expressed with unfused cYFP. Scale bar, 50 μm. (**D**) In vitro phosphorylation of OsWRKY70 by OsMPK3/6. The phosphorylated form of OsWRKY70 (P-WRKY70) was detected by using Phos-tag Biotin BTL-104 (top). Input proteins, including OsWRKY70-His (WRKY70), GST-OsPMK3 (MPK3), GST-OsPMK6 (MPK6), and His-OsMKK4$^{DD}$ (MKK4$^{DD}$) were monitored by Coomassie blue staining. (**E**) Assays for W-box binding activity of OsWRKY70. GST-OsMPK3 or GST-OsMPK6 was activated by a constitutively active form of OsMKK4, His-OsMKK4$^{DD}$. BS65 containing two W-boxes was used as the probe. (**F**) Assays for transactivation activity of OsWRKY70. Leaves of *N. benthamiana* were agroinfiltrated with the indicated constructs. 24 hr later, leaves were injected with 10 mM 17-β-estradiol and were incubated for 12 hr. Total protein was extracted and GUS activities were subsequently quantified. Eight plants were used for each treatment. Letters indicate significant differences among different lines (**A**) or treatments (**F**) ($p < 0.05$, Duncan's multiple range test).

The following figure supplements are available for figure 2:

**Figure supplement 1**. Subcellular localization, DNA-binding ability, and transcriptional activation activity of OsWRKY70.

**Figure supplement 2**. Self-activation of OsWRKY70.

## OsWRKY70 prioritizes defence over growth by regulating phytohormonal signaling

To determine the role of OsWRKY70 in herbivore-induced defence responses, we constructed *OsWRKY70* overexpression and knockdown lines using *Agrobacterium tumefaciens* mediated transformation. Through GUS staining and hygromycin resistance selection, we obtained two homozygous, single-insertion *OsWRKY70*-silenced lines (irWRKY70-7 and irWRKY70-8) and two overexpression lines (oeWRKY70-8 and oeWRKY70-17; *Figure 4—figure supplement 1A*). *OsWRKY70* overexpression resulted in dwarfed plants (*Figure 4B*), suggesting that OsWRKY70 is a negative growth regulator. To reduce phenotypic effects (see below), we also created hemizygous overexpressing lines (hemi-oeWRKY70-8 and hemi-oeWRKY70-17) whose phenotype was weaker but still visible in both nutrient solution and soil (*Figure 4A,B*). SSB-induced transcript levels of *OsWRKY70* in irWRKY70-7 and irWRKY70-8 were suppressed by more than 80% compared to WT plants at 1 hr after SSB feeding. Conversely, *OsWRKY70* transcript levels were increased about 14-fold in hemi-oeWRKY70-8 and hemi-oeWRKY70-17 plants (*Figure 4—figure supplement 1B*). Transcriptional profiling of the *OsWRKY70*-homologous genes *OsWRKY24* and *OsWRKY53* confirmed that gene targeting was specific for *OsWRKY70* (*Figure 4—figure supplement 1C,D*). In soil, the hemi-oeWRKY70 lines displayed dark green leaves and delayed flowering, similar to known GA-deficient mutants (*Sakamoto et al., 2004*). Plant height was reduced by 29% and 27%, and root length by 49% and 30%, respectively (*Figure 4D,E*). In contrast, the irWRKY70 lines grew similar to WT plants (*Figure 4A,B*), except for a slight increase in root length. To test whether *OsWRKY70* acts as a negative regulator of GA biosynthesis, we profiled GA levels in the different lines using HPLC/MS–MS. The experiment revealed that GA$_1$, GA$_7$, GA$_{19}$, GA$_{20}$, GA$_{24}$, and GA$_{53}$ levels were significantly lower in hemi-oeWRKY70 lines (hemi-oeWRKY70-8 and hemi-oeWRKY70-17) than in WT plants (*Figure 4G*). Moreover, the growth phenotype of the oeWRKY70 seedlings was successfully restored to WT levels when they were grown on 1/2 MS plates with GA$_3$ at a concentration of 0.01 μM (*Figure 4C*). Consistently, the GA biosynthesis gene GA 20 oxidase (*GA20ox7*) was significantly down-regulated in the hemi-oeWRKY70-8 lines (*Figure 4F*). These results suggest that OsWRKY70 regulates plant growth through GA biosynthesis.

To understand how OsWRKY70 influences defence signaling in rice, we examined SSB-elicited JA, ET, and SA levels and the expression of biosynthesis genes in *OsWRKY70* transgenic lines and compared them to WT plants. JA levels in the irWRKY70 lines were significantly decreased compared with WT plants upon SSB attack, while they were increased in the overexpressing lines (*Figure 5A*).

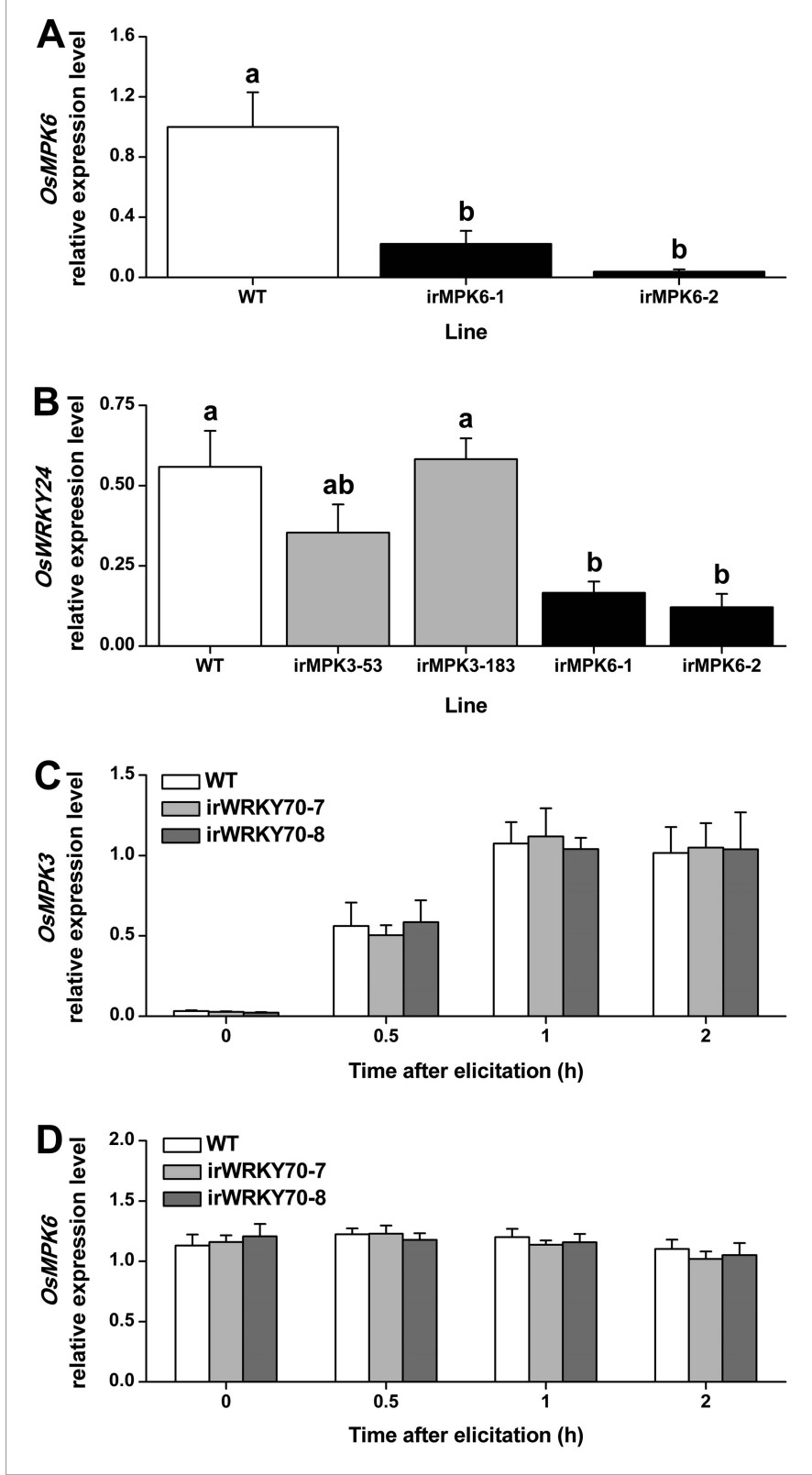

**Figure 3**. Transcript levels of *OsMPK6, OsWRKY24, OsMPK3,* and *OsMPK6* in different transgenic lines. (**A**) Mean expression levels (+SE, n = 6) of *OsMPK6* in *OsMPK6* silenced lines (irMPK6-1 and irMPK6-2). Samples used for QRT-PCR were from plant stems that were infested by SSB for 1 hr. (**B**) Mean transcript levels (+SE, n = 5) of *OsWRKY24* in irMPK3 (irMPK3-53, irMPK3-183) and irMPK6 lines after infestation by SSB for 1 hr. (**C**, **D**) Mean
*Figure 3. continued on next page*

*Figure 3. Continued*

transcript levels (+SE, n = 5) of *OsMPK3* (**C**) and *OsMPK6* (**D**) in irWRKY70 lines after infestation by SSB for 1 hr. Letters indicate significant differences among different lines (p < 0.05, Duncan's multiple range test).

In accordance with this data *OsWRKY70* positively regulated SSB-induced transcript levels of the JA-biosynthesis genes *OsHI-LOX* (*Zhou et al., 2009*) and *OsAOS2* (*Mei et al., 2006*) (*Figure 5B,C*). The accumulation of ethylene was similar in the irWRKY70 lines and WT plants, but the levels were significantly elevated in hemi-oeWRKY70 lines after SSB infestation (*Figure 5D*). Consistent with this result, transcript levels of the ethylene biosynthesis gene *OsACS2* were similar in irWRKY70 and WT plants when infested by SSB, but were much higher in the induced hemi-oeWRKY70 lines (*Figure 5E*). WT plants and ir*WRKY70* lines had nearly identical constitutive and SSB-induced SA levels, whereas the SA levels in the hemi-oe*WRKY70* lines were significantly lower than in WT plants (*Figure 5F*). Isochorismate synthase (ICS) is a key enzyme in plant SA biosynthesis (*Wu and Baldwin, 2010*). We examined the *OsICS1* gene (*Du et al., 2009*) in rice after SSB infestation and found that the *OsICS1* transcriptional level was significantly decreased in the hemi-oe*WRKY70* lines compared with WT plants (*Figure 5G*). Taken together, these experiments demonstrate that OsWRKY70 positively regulates SSB-induced JA- and ET levels but negatively regulates SA levels. To explore the notion that OsWRKY70 may be an upstream regulator of the JA and ET pathways, we investigated the expression of *OsWRKY70* in transgenic plants with impaired JA and ET signaling. We used an antisense *OsHI-LOX* line (as-*lox*), which produces 50% less JA upon SSB infestation than WT plants (*Zhou et al., 2009*), and an antisense-*ACS2* line (as-*acs*), which produces significantly less SSB-elicited ET than WT plants (*Lu et al., 2011*). The experiments revealed that the levels of constitutive and SSB-induced *OsWRKY70* transcripts in as-*lox* and as-*acs* plants were the same as those in WT plants over the first 60 min of infestation (*Figure 6A,B*), suggesting that OsWRKY70 functions upstream of JA and ET signaling. To fully demonstrate that OsWRKY70 acts upstream of JA and ET, additional experiments with null-mutants would be required.

## OsWRKY70-dependent defence prioritization increases resistance to a chewing herbivore through JA-dependent defense activation

Trypsin protease inhibitors (TrypPIs) are important direct defence proteins against SSB in rice and their activity is regulated by JA- and ET (*Zhou et al., 2009*; *Li et al., 2012*). Thus, we investigated the influence of OsWRKY70 on TrypPI activity and SSB performance. TrypPI activity was suppressed in the irWRKY70 lines and enhanced in the hemi-oeWRKY70 lines compared with WT plants (*Figure 7A*). As expected, SSB caterpillars gained more mass on irWRKY70-7 and irWRKY70-8 plants and less mass on the overexpressing lines compared to those fed on WT plants (*Figure 7B*). IrWRKY70 lines were more severely damaged by SSB than the WT plants, whereas the hemi-oeWRKY70 lines were less damaged (*Figure 7C,D*). To determine whether the impaired SSB resistance and defences in the irWRKY70 lines is due to lower JA levels, we complemented irWRKY70 plants with JA and examined SSB-induced TrypPI production and SSB performance. JA treatment attenuated the difference in TrypPI levels between WT plants and the irWRKY70 lines (*Figure 7E*). Moreover, SSB larvae fed on JA-treated irWRKY70 lines gained a similar amount of weight to those fed on equally treated WT plants (*Figure 7F*). The complete restoration of plant resistance to SSB and elicited accumulation of TrypPIs in irWRKY70 by exogenous JA application suggests that OsWRKY70 mediates rice-resistance to SSB through JA signaling.

## OsWRKY70 dependent, GA-mediated growth suppression increases susceptibility to a non-target herbivore

Based on the above results, we investigated whether OsWRKY70 regulation influences plant resistance to a non-target herbivore (i.e., a secondary attacker that does not strongly activate OsWRKY70): the piercing sucking rice BPH *N. lugens*. When irWRKY70 lines and WT plants were exposed to a BPH colony, adult females preferred feeding on the WT rather than the irWRKY70 lines (*Figure 8A,B*). Similarly, BPH adult females laid more eggs on WT plants than irWRKY70 (*Figure 8A,B*, inserts). In accordance with these findings, BPH adult females were found more often on hemi-oeWRKY70 lines than on WT plants

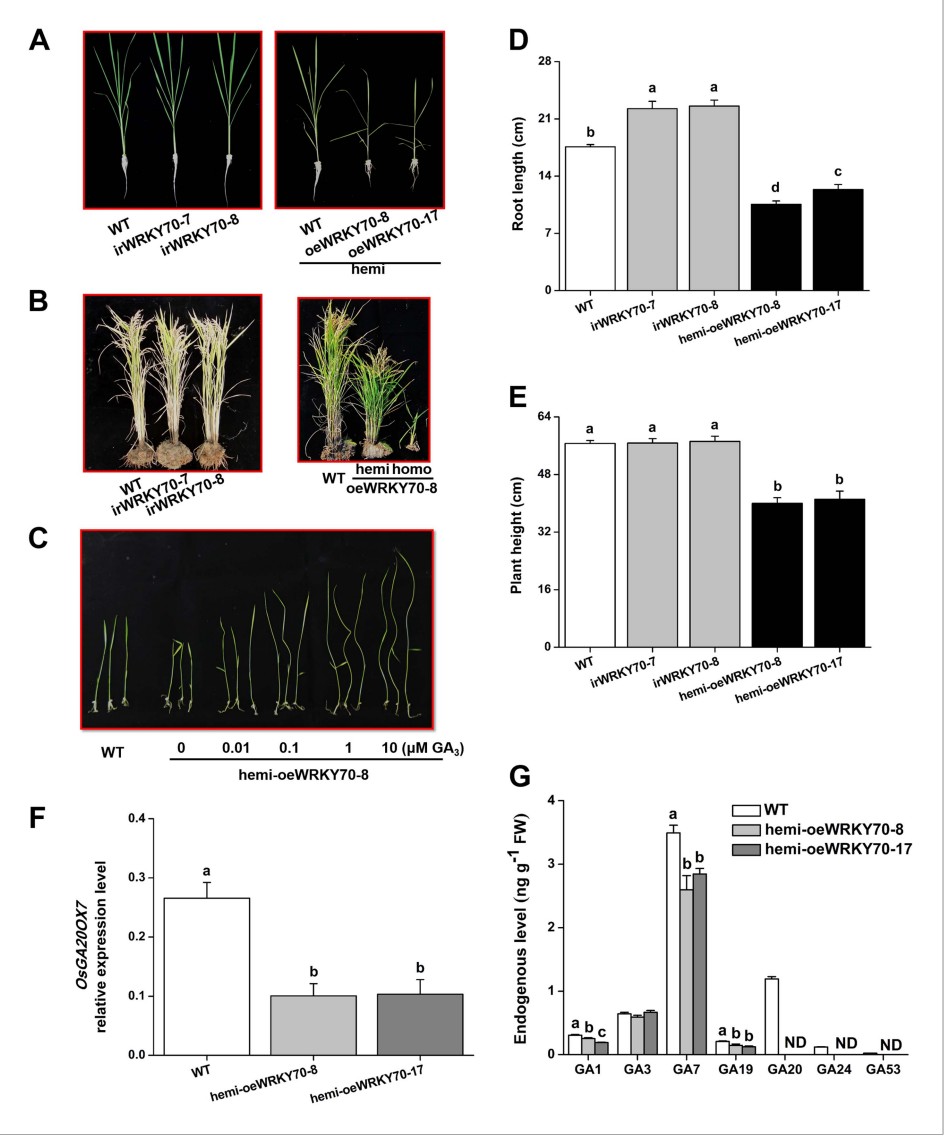

**Figure 4**. Altering OsWRKY70 expression affects GA levels and plant growth. (**A**, **B**) Growth phenotypes of OsWRKY70 transgene lines (irWRKY70 lines, irWRKY70-7 and irWRKY70-8, and oeWRKY70 and hemi-oeWRKY70 lines, oeWRKY70-8, hemi-oeWRKY70-8 and hemi-oeWRKY70-17) and wild-type (WT) plants at tillering stage (**A**) and heading stage (**B**). (**C**) 10-day-old seedlings of WT and hemi-oeWRKY70-8 lines whose seeds were surface sterilized and placed on 1/2 Murashige and Skoog agar medium containing $GA_3$ (minimum purity > 99%, Sigma, St Louis, MO) at various concentrations. This experiment was repeated 3 times with similar results. (**D**, **E**) Root length (**D**) and plant height (**E**) of transgenic lines with silencing (irWRKY70) or overexpressing (hemizygous lines, hemi-oeWRKY70 lines) of *OsWRKY70* and WT plants at tillering stage. (**F**) Mean transcript levels (+SE, n = 5) of *OsGA20ox7* in hemi-oeWRKY70-8, hemi-oeWRKY70-17, and WT plant. (**G**) Mean levels (+SE, n = 3) of gibberellins (GAs), including $GA_1$, $GA_3$, $GA_7$, $GA_{19}$, $GA_{20}$, $GA_{24}$, and $GA_{53}$, in hemi-oeWRKY70-8, hemi-oeWRKY70-17, and WT plants. Letters indicate significant differences among different lines (p < 0.05, Duncan's multiple range test).

The following figure supplements are available for figure 4:

**Figure supplement 1**. *OsWRKY70* transgenic lines and levels of *OsWRKY70, OsWRKY24*, and *OsWRKY53* transcripts in the transgenic lines and WT plants.

**Figure supplement 2**. Elongation of the second leaf sheath in hemi-oeWRKY70-8 and WT plants in response to $GA_3$.

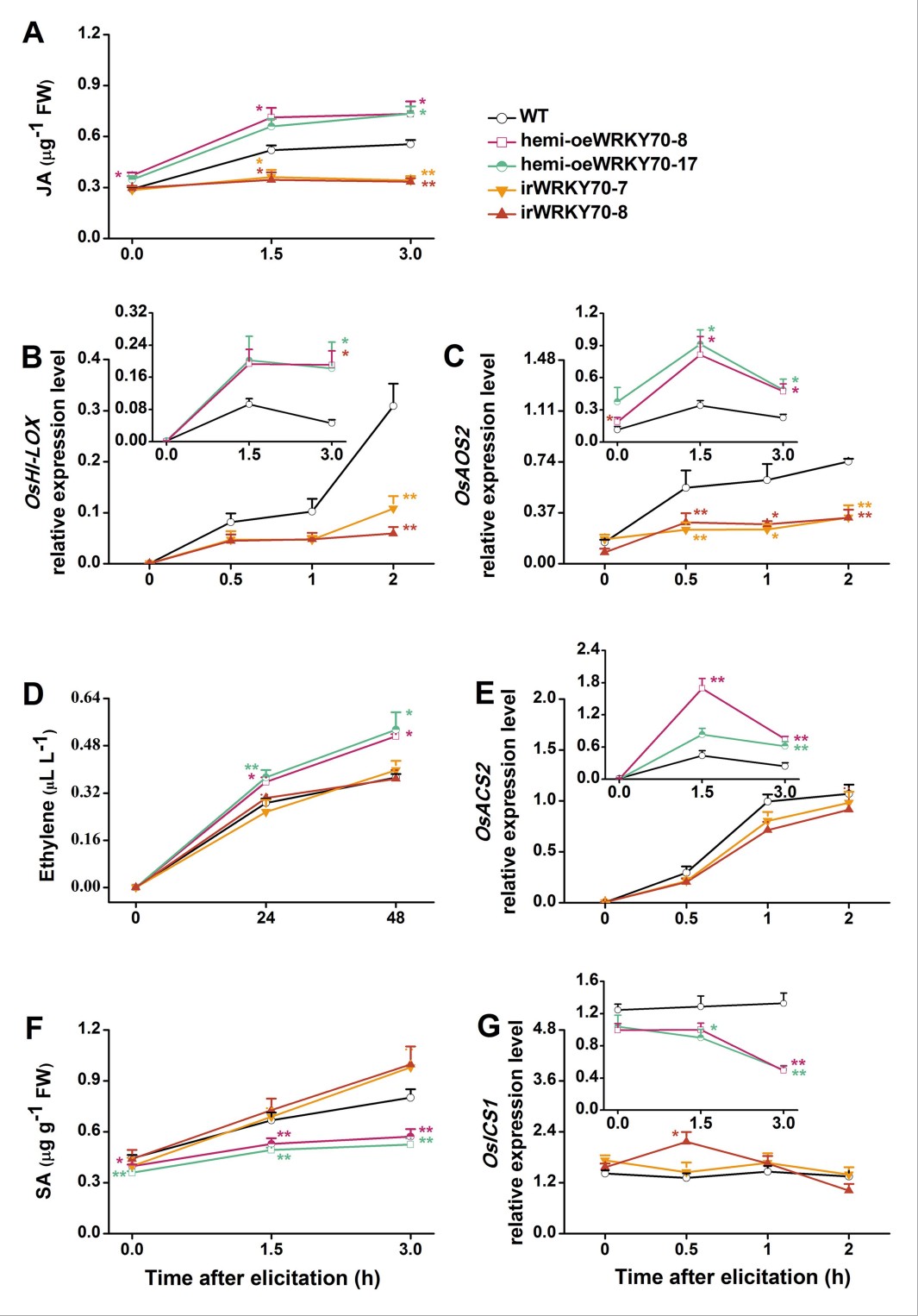

**Figure 5**. OsWRKY70 mediates SSB-elicited JA, SA, and ET accumulation. Mean levels (+SE, n = 5–10) of JA (**A**), ET (**D**), and SA (**F**), and mean expression levels (+SE, n = 5) of *OsHI-LOX* (**B**), *OsAOS2* (**C**), *OsACS2* (**E**), and *OsICS1* (**G**) in irWRKY70, hemi-oeWRKY70, and WT plants that were individually infested by a third-instar SSB larva. Asterisks
*Figure 5. continued on next page*

*Figure 5. Continued*

indicate significant differences in irWRKY70, hemi-oeWRKY70 compared with WT plants (*, p < 0.05; **, p < 0.01; Duncan's multiple range test).

The following figure supplement is available for figure 5:

**Figure supplement 1**. W-box elements in promoter regions of *OsHI-LOX, OsICS1, OsAOS2, OsACS2,* and *OsGA20ox7.*

and laid more eggs on the former than on the latter (*Figure 8C,D*). Moreover, BPH nymphs fed on the irWRKY70 lines had lower survival rates than those fed on WT plants; in contrast, BPH nymphs fed on the hemi-oeWRKY70 lines had higher survival rates (*Figure 8E,F*), showing that OsWRKY70 negatively regulates rice BPH resistance. Based on the signaling profiles showing that OsWRKY70 negatively regulates GAs, we hypothesized that the down regulation of GAs may be responsible for the enhanced susceptibility to BPH. We therefore conducted a series of experiments to explore the influence of OsWRKY70 dependent GA on BPH. First, we used a GA-deficient mutant, *sd-1* (*Spielmeyer et al., 2002*), and a GA-excessive mutant, *eui* (*Zhu et al., 2006*), to test the importance of GA for BPH resistance. The BPH female adults preferred feeding and ovipositing on *sd-1* rather than the WT (ZH11) (*Figure 9A*), whereas the *eui* lines repelled BPH feeding compared with WT plants and did not affect BPH oviposition (*Figure 9B*). The BPH nymph mortality was significant higher on the *eui* mutant compared with WT plants, but the *sd-1* mutant did not influence BPH nymph performance (*Figure 9C*). Second, we complemented the hemi-oeWRKY70 lines with GA$_3$ at a concentration of 1 µM. This treatment restored BPH resistance to WT levels: BPH female adults feeding and ovipositing showed no preference between the WT and GA$_3$-treated hemi-oeWRKY70 lines (*Figure 9D,E*), and the BPH nymph survival rate was the same on GA$_3$-treated hemi-oeWRKY70 and WT plants (*Figure 9F*). Taken together, these results strongly suggest that OsWRKY70 negatively regulates BPH resistance through GA signaling.

## Discussion

Our experiments demonstrate that prioritizing defence over growth in response to a chewing herbivore is linked to a trade-off with resistance against a piercing-sucking herbivore via a WRKY TF. This indirect additional cost of defence may lead to the evolution of divergent, herbivore-community dependent plant resistance strategies in nature and may significantly constrain efforts to breed herbivore-resistant plants. It has been well documented that there are trade-offs between plant growth and defence (*Zavala and Baldwin, 2006*; *Zhang et al., 2008*; *Meldau et al., 2012*; *Yang et al., 2012*). Resource availability, competition, plant ontogeny, and herbivory can influence the allocation of resources to growth and defence (*Stamp, 2003*; *Boege and Marquis, 2005*). In nature, defence prioritization is complicated by the fact that plants are often attacked simultaneously by multiple herbivore species, which have different sensitivities to various defence strategies, leading to resistance trade-offs (*Stam et al., 2014*). For example, leaf-chewing caterpillars were found to perform better on *Arabidopsis* plants that are attacked by phloem-sucking aphids and *vice versa* (*Soler et al., 2012*). Given that herbivory is an important driving force for the evolution of plant defence (*Agrawal et al., 2012*; *Züst et al., 2012*), understanding growth/defence and resistance trade-offs is important to predict and understand selection patterns in nature. Our study reveals that growth/defence and resistance trade-offs can emanate from the same mechanistic basis. From a plant's perspective, this suggests that reducing growth to support defence is even more costly than previously anticipated. From an agricultural point of view, this result indicates that it may be problematic to breed resistant varieties that rely on induced defence, as these plants may suffer from both a depression in growth and increased susceptibility to non-target herbivores.

The discovery and manipulation of a TF that directly regulates defence prioritization allows us to draw a detailed picture of the mechanisms that underlie defence prioritization in rice. OsWRKY70 is rapidly induced following mechanical wounding and SSB feeding, but not following attack by a piercing sucking herbivore. Despite a lacking D-domain, OsWRKY70 interacts with and is regulated by two MAP-kinases, OsMPK3 and OsMPK6 (*Figure 2*). It has been well documented that Group I-type WRKY TFs can be phosphorylated by MAPKs and that the SP clusters are the phosphorylating sites

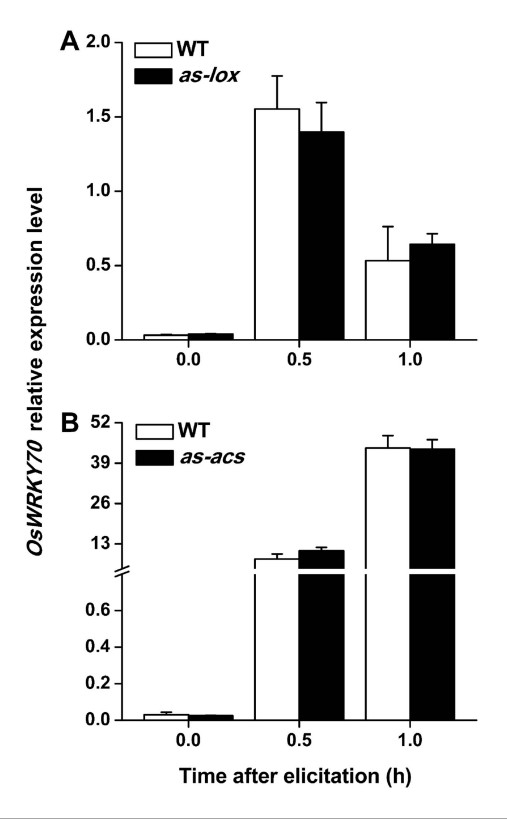

**Figure 6**. Levels of *OsWRKY70* transcripts in WT plants and transgenic lines with impaired JA (as-*lox*) and ethylene (as-*acs*) biosynthesis. Mean transcript levels (+SE, n = 5) of *OsWRKY70* in transgenic lines with impaired JA (**A**, as-*lox*) and ethylene (**B**, as-*acs*) bio-synthesis and WT plants after they were infested by SSB.

(*Mao et al., 2011*; *Ishihama and Yoshioka, 2012*). We found that both OsMPK3 and OsMPK6 phosphorylate OsWRKY70, which in turn increased the transactivation activity of OsWRKY70 (*Figure 2*). Moreover, OsWRKY70 can auto-regulate itself (*Figure 2—figure supplement 2*). Given that autoregulation and cross-regulation are common features of WRKY action (*Ishihama and Yoshioka, 2012*), OsWRKY70 transcript levels are likely reduced in OsMPK3 and 6 silenced lines because of the reduction in phosphorylated WRKY70 and other WRKYs, which decreases WRKY activity and thereby reduce OsWRKY70 transcript levels. Phytohormones on the other hand do not regulate OsWRKY70. Combined with its capacity to bind to W-box sequences and to act as transcriptional activator, this places OsWRKY70 at the interface between early recognition and signaling and hormonal regulation (*Rushton et al., 2010*; *Wu and Baldwin, 2010*; *Erb et al., 2012*). Indeed, silencing and over-expression of *OsWRKY70* demonstrates its central role in regulating defence and growth through JA, ET, SA, and GA signaling, which enables OsWRKY70 to reduce plant growth and increases defence upon herbivore attack. In other plant systems, WRKYs have also been reported to play important roles in the regulation of transcriptional reprogramming associated with plant growth, development, and stress responses at different levels, including upstream and downstream of protein kinases and hormones (*Rushton et al., 2010*; *Ishihama et al., 2011*; *Mao et al., 2011*; *Zheng et al., 2013*). NaWRKY3 and NaWRKY6, the homologs of OsWRKY70 in *N. attenuata*, for instance, have been reported to function downstream of NaSIPK and NaWIPK and upstream of JA biosynthesis (*Skibbe et al., 2008*). In *Arabidopsis*, AtWRKY33, the homolog of OsWRKY70 is phosphorylated by AtMPK3/MPK6 and can affect ET synthesis by directly binding promoter region of *ACS2* and *ACS6* genes (*Li et al., 2012*). Another rice WRKY, OsWRKY24 has been reported to repress GA signaling in rice aleurone cells (*Zhang et al., 2009*).

OsWRKY70 may regulate phytohormone signaling via two, not mutually exclusive routes. First, it may directly bind to genes that are involved in hormone biosynthesis and signaling. Consistent with this hypothesis, OsWRKY70 positively modulated the transcript levels of the JA- and ET-synthesis genes *OsHI-LOX*, *OsAOS2*, and *OsACS2*, and negatively regulated the transcripts of the SA- and GA-biosynthesis genes *OsICS1* and *OsGA20ox7* (*Figures 4F*, *5B,C,E,G*). The existence of W-box or W-box like motifs in the promoters of these genes (*Figure 5—figure supplement 1*) provides additional indirect evidence for their interaction with OsWRKY70. Second, OsWRKY may regulate growth and defence through indirect hormonal cross-talk. It has been reported that the JA-signaling pathway is connected to the GA-signaling pathway through COI1-JAZ1-DELLA-PIF complexes, resulting in mutual suppression. The activation of JA signaling inhibits GA-mediated plant growth, whereas the activation of the GA pathway inhibits JA-mediated plant defence (*Yang et al., 2012*). Moreover, JA was found to inhibit GA biosynthesis via an unknown mechanism (*Yang et al., 2012*; *Heinrich et al., 2013*), and the GA-GID1-DELLA complex was found to positively regulate the production of SA (*Navarro et al., 2008*). Thus, the observed phytohormone levels and associated phenotypes in the transgenic OsWRKY70 lines might be at least in part due to antagonistic and synergistic phytohormone crosstalk. It has also been reported that the absence of JA signaling enhances the sensitivity of plants to GAs (*Yang et al., 2012*). In our experiments, the promotion of JA signaling in oeWRKY70 lines did not decrease the sensitivity of

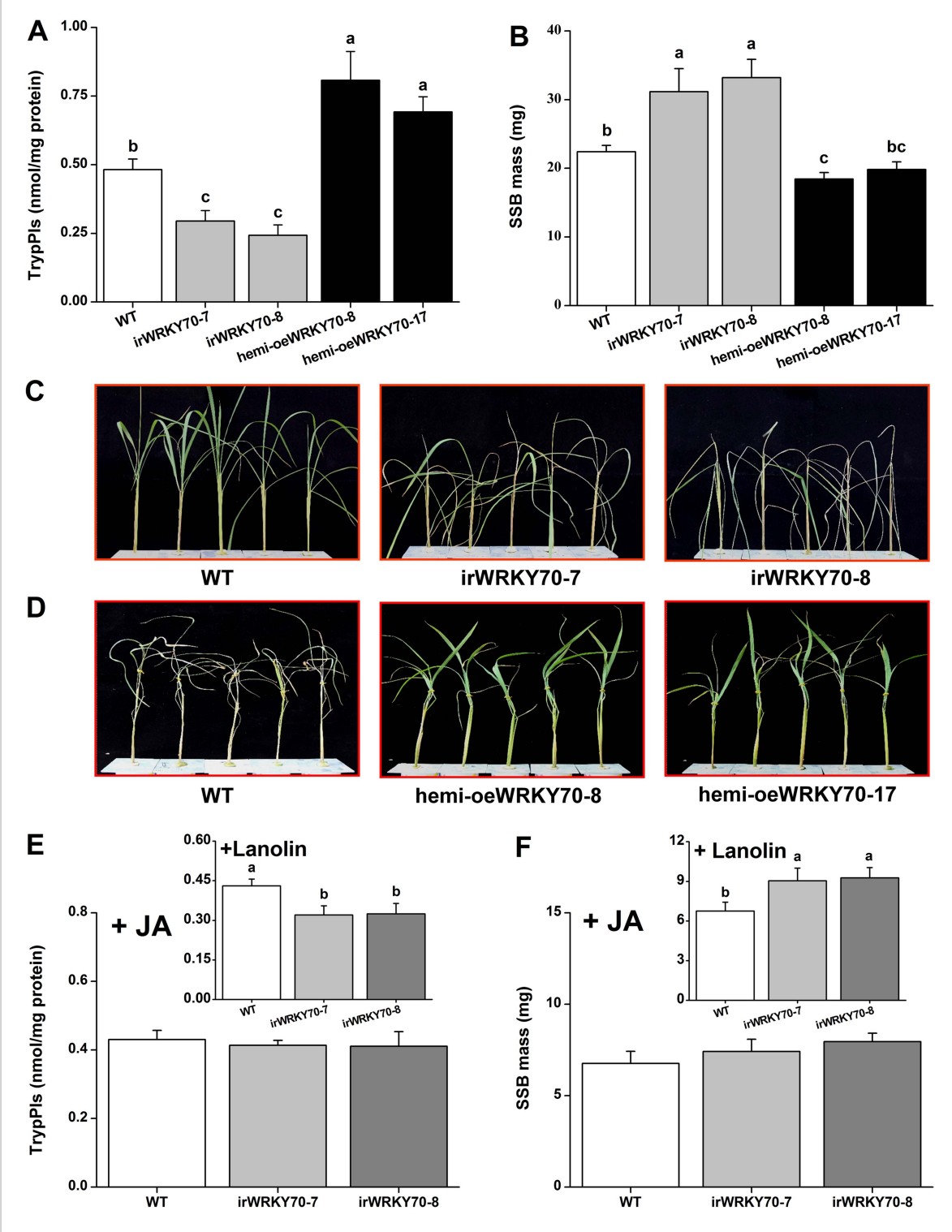

**Figure 7**. OsWRKY70 positively regulates resistance in rice to SSB. (**A**) Mean Trypsin protease inhibitor (TrypPI) activities (+SE, n = 6) in irWRKY70, hemi-oeWRKY70, and WT plants that were individually infested by a third-instar SSB larva for 3 days. (**B**) Mean larval mass (+SE, n = 50) of SSB that fed on irWRKY70, hemi-oeWRKY70, and WT plants for 14 days. (**C**, **D**) Damaged phenotypes of irWRKY70 (**C**), hemi-oeWRKY70 (**D**), and WT plants that were individually infested by a third-instar SSB larva for 8 days (n = 10). This experiment was repeated twice with similar results. (**E**) Mean activities (+SE, n = 6) of TrypPIs in irWRKY70 and WT plants that were individually treated either 100 µg JA in 20 µl of lanolin paste (JA) or with 20 µl of pure lanolin (insert) for 24 hr, followed by SSB feeding for 3 days; (**F**) Mean larval mass (+SE, n = 50) of SSB 12 days after fed on irWRKY70 and WT plants that were individually treated either 100 µg JA in 20 µl of lanolin paste (JA) or with 20 µl of pure lanolin (insert) for 24 hr. Letters indicate significant differences among different lines (p < 0.05, Duncan's multiple range test).

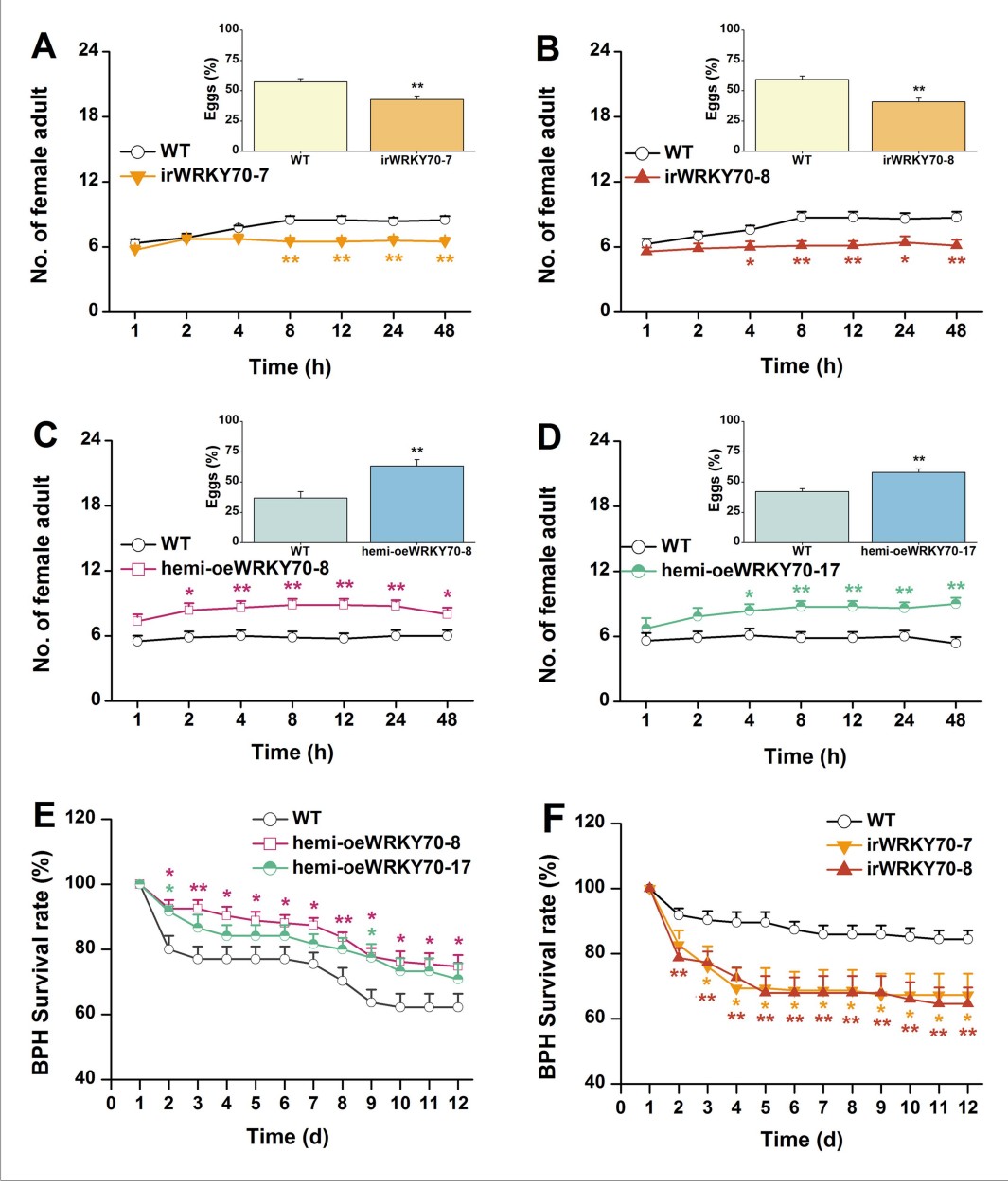

**Figure 8.** OsWRKY70 negatively regulates resistance of rice to BPH. (**A–D**) Mean number of female BPH adults per plant (+SE, n = 8) on pairs of plants (WT vs irWRKY70-7, irWRKY70-8, hemi-oeWRKY70-8, and hemi-oeWRKY70-17, respectively), 1–48 hr after pairs were exposed. Inserts: mean percentage (+SE, n = 8) of BPH eggs per plant on pairs of plants as started above, 48 hr after the release of BPH. (**E**, **F**) Mean survival rate (+SE, n = 10) of BPH nymphs that fed on irWRKY70, hemi-oeWRKY70, or WT plants 1–12 days after the start of feeding. Asterisks indicate significant differences in irWRKY70, hemi-oeWRKY70 compared with WT plants (*, $p < 0.05$; **, $p < 0.01$; Student's *t*-test [**A-D**] or Duncan's multiple range test [**E**, **F**]).

plants to $GA_3$ (*Figure 4—figure supplement 2*). Moreover, the dwarf phenotype of oeWRKY70 lines was completely restored by exogenous $GA_3$ at low concentrations of 0.01 μM (*Figure 4C*). This suggests that the dwarf phenotype of oeWRKY70 lines is directly related to the low level of GAs and that the sensitivity of plants to GAs may be influenced by other *OsWRKY70*-mediated factors other than JA. Interestingly, irWRKY70 lines showed similar growth phenotypes to WT plants, indicating that GA levels are unlikely to be altered in irWRKY70 lines. This suggests an involvement of other factors, such as the homologs of OsWRKY70, OsWRKY24, and OsWRKY53, in the biosynthesis of GAs. Overall, the

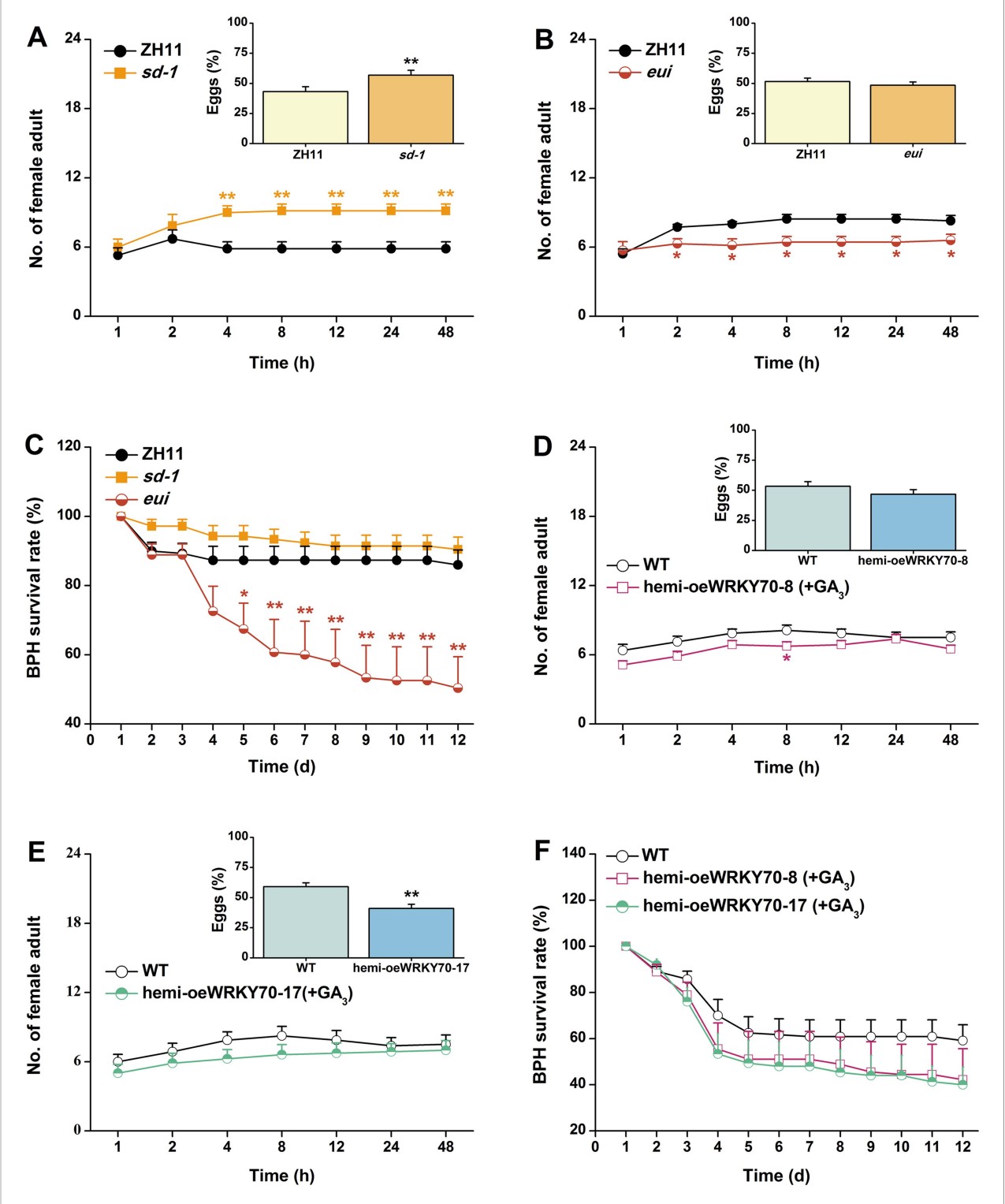

**Figure 9.** The GA-signaling pathway positively regulates rice resistance to BPH. (**A**, **B**) Mean number of adult female BPH per plant (+SE, n = 8) on pairs of plants (WT (ZH11) vs *sd-1* and *eui*, respectively), 1–48 hr after pairs were exposed. Inserts: mean percentage (+SE, n = 8) of BPH eggs per plant on pairs of plants as started above, 48 hr after the release of BPH. (**C**) Mean survival rate (+SE, n = 10) of BPH nymphs that fed on *sd-1*, *eui* lines, or WT (ZH11) plants 1–12 days after the start of feeding. (**D**, **E**) Mean number of female BPH adults per plant (+SE, n = 8) on pairs of plants, a WT plant that was grown in a nutrient solution without GA$_3$ vs a hemi-oeWRKY70-8 (**D**) or hemi-oeWRKY70-17 (**E**) plant that was grown in a nutrient solution with GA$_3$ at a concentration of 1 μM for 24 hr, 1–48 hr after pairs were exposed. Inserts: mean percentage (+SE, n = 8) of BPH eggs per plant on pairs of plants as started above, 48 hr after the release of BPH. (**F**) Mean survival rate (+SE, n = 10) of BPH nymphs that fed on WT plants that were grown in a nutrient

*Figure 9. continued on next page*

Figure 9. Continued

solution without GA$_3$ or hemi-oeWRKY70 lines (hemi-oeWRKY70-8 and hemi-oeWRKY70-17) that had been grown in a nutrient solution with GA$_3$ at a concentration of 1 μM for 24 hr, 1–12 days after the start of feeding. Asterisks indicate significant differences in mutants compared with WT plants (*, $p < 0.05$; **, $p < 0.01$; Student's $t$-test [**A**, **B**, **D**, **E**] or Duncan's multiple range test [**C**]).

combination of direct regulation and indirect phytohormone crosstalk may explain how a single TF can act as a node in multiple signaling processes and integrate growth and defence responses.

Our experiments do not only connect OsWRKY70 with phytohormone signaling, but also illustrate how hormonal signaling affects resistance responses against different herbivores. In rice, it is well documented that TrypPIs are an effective JA-dependent defence against chewing herbivores, including SSB (*Zhou et al., 2009*; *Lu et al., 2011*). Here, we found that OsWRKY70 positively mediated the production of elicited JA and TrypPIs (*Figures 5A*, *7A*), which subsequently modulated resistance in rice to SSB (*Figure 7B–D*). Moreover, JA complementation of irWRKY70 lines, which had lower herbivore-elicited JA levels, completely restored TrypPI activity and SSB resistance compared to equally treated WT plants (*Figure 7E,F*). These data suggest that OsWRKY70-mediated resistance in rice to SSB is mainly due to its effect on the JA-signaling pathway. On the other hand, little is known so far about the role of GA in herbivore resistance against piercing-sucking insects. We found that GA$_3$ application restored BPH resistance in hemi-oeWRKY70 mutants (*Figure 9D–F*). Moreover, the GA-deficient mutant *sd-1* improved the performance of BPH, whereas the GA-excessive mutant *eui* decreased BPH performance (*Figure 9A–C*). These data demonstrate that the GA-signaling pathway plays an important role in modulating resistance in rice to BPH in addition to its regulation of plant growth. GA-modulated BPH resistance in rice may occur via two mechanisms. One is that GA directly regulates the BPH defence response. It has been reported that GA can positively modulate the pathogen-related *PBZ1* gene (*Tanaka et al., 2006*) and cell modification (*Yang et al., 2008*). The rigidity of the cell wall is important for resistance to BPH phloem-feeding. Moreover, GA can directly elicit the plant growth, which may enhance the tolerance of rice to BPH. Another possibility is that GA indirectly regulates BPH resistance by eliciting SA and ROS pathways, both of which have been reported to be involved in resistance of rice to BPH (*Zhou et al., 2009*; *Lu et al., 2011*), via GA-GID1-DELLA complex (*Navarro et al., 2008*; *Alonso-Ramírez et al., 2009*). Thus far, several other elements of the rice defense signaling cascade, including JA (*Zhou et al., 2009*), OsERF3 (*Lu et al., 2011*), and ethylene (*Lu et al., 2014*) have also been shown to have similar divergent effects on SSB and BPH, suggesting distinct resistance strategies of rice plants against these two herbivores. The main reason for the divergent signaling, response and resistance to the two herbivores might be their different feeding habits. Resistance mechanisms against phloem-feeders are well documented to differ substantially from mechanisms against chewing herbivores (*Bostock, 2005*).

In summary, our experiments provide evidence for a key role of OsWRKY70 in defence prioritization and illustrate that reducing growth via GA signaling opens the door to secondary infection by non-target herbivores. When attacked by a chewing herbivore such as SSB, rice plants will recognize signals derived from the herbivore and activate OsMPK3 and OsMPK6. The activated OsMPK3 and OsMPK6 then elicit OsWRKY70, which subsequently activates the JA- and ET-signaling pathways, resulting in the production of defence compounds such as TrypPIs and an increase in plant resistance. Simultaneously, the activation of OsWRKY70 decreases the production of GAs, which inhibits plant growth and thus prioritizes defence overgrowth and leads to increased susceptibility to BPH. Through these results, our study illustrates that the transcriptional modulation of hormonal networks allows plants to mount an appropriate defence program. At the same time, however, prioritizing defence over growth leads to significant resistance trade-offs, which may constrain plant resistance breeding and favor the evolution of herbivore-specific responses in plants.

## Materials and methods

### Plant growth and insects

The following rice genotypes were used in the present study: (i) Xiushui 11 WT plants and the corresponding transgenic lines irWRKY70, hemi-oeWRKY70 (see below), as-*lox* (*Zhou et al., 2009*), as-*acs* (*Lu et al., 2011*), irMPK3 (*Wang et al., 2013*), irMPK6 (*Figure 4A*) and (ii) Zhonghua 11 (ZH11) WT plants and the corresponding GA mutants *sd-1* (*Spielmeyer et al., 2002*) and *eui* (*Zhu et al., 2006*).

Pre-germinated seeds were cultured in plastic bottles (diameter 8 cm, height 10 cm) in a greenhouse ($28 \pm 2°C$, 14L: 10D). 10-day-old seedlings were transferred to 50 L hydroponic boxes with a rice nutrient solution (*Yoshida et al., 1976*). After 30–35 days, seedlings were transferred to individual 500 ml hydroponic plastic pots. Plants were used for experiments 4–5 days after transplantation. A colony of SSB was originally obtained from rice fields in Hangzhou, China and maintained on TN1 rice seedlings using the method described previously (*Zhou et al., 2009*). For BPH, we used a lab population that has been reared on TN1 rice seedlings for more than 20 generations.

## Isolation of *OsWRKY70* cDNA

The full-length cDNA of *OsWRKY70* was obtained by RT-PCR from total RNA isolated from WT plants infested by SSB larvae for 24 hr. The primers were designed based on the sequence of the rice *OsWRKY70* (TIGR ID Os05g39720) gene (*Supplementary file 1A*), which showed high homology with the partial sequence of the OsWRKY70 transcript that was cloned by SSH. The PCR-amplified fragments were cloned into the pMD 19-T vector (TaKaRa, China) (pOsWRKY70) and sequenced.

## Generation and characterization of transgenic plants

The full-length cDNA and a 443 bp fragment (*Figure 1—figure supplement 1*) of *OsWRKY70* were cloned into the pCAMBIA1301 and pCAMBIA1301-RNAi vectors, respectively, yielding an over-expression (oeWRKY70) and an inverted-repeat orientation (irWRKY70) vector. Both the oeWRKY70 and irWRKY70 vectors were inserted into the rice variety Xiushui 11 using *A. tumefaciens*-mediated transformation. Homozygous $T_2$ plants were selected using GUS staining or hygromycin resistance screening (*Zhou et al., 2009*). For most experiments, two irWRKY70 T2 homozygous lines, irWRKY70-7 and irWRKY70-8, each harboring a single insertion were used. However, oeWRKY70 homozygous lines were severe dwarfing and nearly no seeds, thus we used two hemizygous lines, hemi-oeWRKY70-8 and hemi-oeWRKY70-17, each also harboring a single insertion to perform the experiments.

## Plant treatments

### Mechanical wounding

Plants (one per pot) were individually damaged using a needle on the lower part of the stems (about 2 cm long), with 200 holes (W). Control plants (Control) were not pierced.

### SSB treatment

Plants (one per pot) were individually infested using a third-instar SSB larva that had been starved for 2 hr. Control plants (Control) were left herbivore-free.

### BPH treatment

Plants (one per pot) were individually infested with 15 female BPH adults that were confined in a glass cage (diameter 4 cm, height 8 cm, with 48 small holes, diameter 0.8 mm). Plants with an empty cage were used as controls (non-infested).

### JA and SA treatment

The method for JA and SA treatment was the same as described previously (*Zhou et al., 2009*). Plants were individually sprayed with 2 ml of JA (100 μg/ml) or SA (70 μg/ml) in 50 mM sodium phosphate buffer. Control plants were sprayed with 2 ml of the buffer (Buffer). For JA complementation experiment, ir*WRKY70* line stems were individually treated with 100 μg of JA in 20 μl of lanolin paste. Controls (Lanolin) were similarly treated with 20 μl of pure lanolin.

### $GA_3$ treatment

For hemi-oeWRKY70 line growth complementation experiment, plants grown in one-half Murashige–Skoog (MS) medium with 0.4% phytogel supplemented with $GA_3$ (minimum purity > 99%, Sigma, St Louis, MO) at various concentrations (see details in *Figure 4*). The growth phenotype was observed 10 days later. For BPH resistance complementation experiments, individual rice seedlings were grown in a nutrient solution (pH 4.8) with $GA_3$ at a concentration of 1 μM (*Li et al., 2011*). Plants grown in nutrient solution without $GA_3$ were used as controls.

## Expression and purification of recombinant protein

The full-length ORF of *OsWRKY70* was PCR-amplified and cloned into the pET-32a vector (Novagen, Madison, WI). The full-length ORF of *OsMKK4* was PCR-amplified and cloned into the pET-28b vector

(Novagen), and the two phosphorylation sites (T and S) of OsMKK4 were mutant to D (OsMKK4$^{DD}$) by Q5 Site-Directed Mutagenesis Kit (NEB). The full-length ORFs of *OsMPK3* and *OsMPK6* were PCR-amplified and cloned into the pGEX-4T-3 vector (GE Healthcare). All of primers used for PCR amplification for these genes are listed in *Supplementary file 1A*. The constructs were transformed into *E. coli* BL21 (DE3) (Transgene, China). Expression was induced by adding 0.4 (for *OsWRKY70* and *OsMKK4$^{DD}$*) or 0.2 (for *OsMPK3* or *6*) mM isopropyl-β-thiogalactopyranoside (IPTG) for 20 hr at 20°C (for *OsWRKY70* and *OsMKK4$^{DD}$*) or for 4 hr at 23°C (for *OsMPK3* and *6*). Cells were collected and the recombinant protein was purified using His or GST Trap (GE healthcare, UK) according to the manufacturer's instructions.

## Yeast one hybrid assay

The full-length ORF of *OsWRKY70* was PCR-amplified and cloned into the GAL4 DNA-binding domain of the pGBKT7 vector (Clontech, Palo Alto, CA). The vector construct was transformed into yeast Y187 (Clontech) according to the manufacturer's instructions. Transformants were selected on SD (−Trp) plates at 30°C until colonies appeared. The colonies were identified by PCR and transferred into SD (−Trp) liquid medium. The transformant yeast with pGBKT7 or pGBKT7-OsWRKY70 was plated on SD (−Trp) containing X-α-gal at 30°C for 12 hr until the pGBKT7-OsWRKY70 transformants developed a blue color.

## Subcellular localization of OsWRKY70

The full-length ORF without a stop codon of *OsWRKY70* was cloned into the pEGFP vector (Clontech) to fuse it with GFP. The fusion gene, *OsWRKY70:GFP*, was inserted into pCAMBIA1301, yielding a transformation vector. This vector was used for transient transformation of *N. benthamiana* leaves as described previously (*Li et al., 2014*). Fluorescence was analyzed by confocal microscopy.

## BiFC assays

Full-length ORFs of OsMPK3, OsMPK6, and OsWRKY70 without stop codons were cloned into serial pGreen-pSAT1 vectors containing either amino- or carboxyl terminal EYFP fragments and introduced into *Agrobacterium* as described previously (*Hou et al., 2010*). 3-week-old *N. benthamiana* leaves were agroinfiltrated with agrobacterial cells containing the indicated constructs. 2 days after incubation, fluorescence and DAPI staining were analyzed by confocal microscopy.

## In vitro pull-down assay

Pull-down assay was performed as described previously (*Ishihama et al., 2011*); 5 μg of GST-tagged OsMPK3 and OsMPK6 and 2 μg His$_6$-tagged OsWRKY70 were used. The samples were analyzed by SDS-PAGE. After electrophoresis, the gels were stained with Coomassie Brilliant Blue or subjected to immunoblot analysis using anti-WRKY70 antibody. Antibody with specificity to OsWRKY70 was generated by immunizing rabbits with the peptide CVYYASRAKDEPRDD-keyhole limpet hemocyanin-conjugate, and purified by the GenScript Company (Nanjing, China).

## Transactivation activity assay

The full-length ORFs of *OsMPK3*, *OsMPK6*, and *OsMKK4$^{DD}$* without stop codons were cloned into pBA-YFP, pBA-YFP, and 6myc-pBA, respectively. 1.8-kb promoter region of OsWRKY70 was PCR-amplified (primers were listed in *Supplementary file 1A*) and cloned into pCAMBIA1391. All constructs were introduced into AGL1 *Agrobacterium*. Leaves of *N. benthamiana* were agroinfiltrated with the indicated constructs (see details in *Figure 2E* and *Figure 2—figure supplement 2C*) at a ratio of 1:1:1:1. At 24 hr after agroinfiltration, leaves were injected with 10 mM 17-β-estradiol and were incubated for 12 hr. 2 days after infiltration, leaves were harvested and frozen in liquid nitrogen. Each treatment was repeated 8 times. GUS quantitative assay was performed as described (*Xin et al., 2012*).

## Electrophoretic mobility shift assay (EMSA)

The probes used in EMSA were BS65 (5′-ATCGTTGACCGAGTTGACTTT-3′) with two W-boxes, P70 (GCCAGTCAAACCTCGAGGGAGCTTTGACCAGTCAACGGTCAAACGTTCAAAGGTCTATATAATGA TCACCGGAGGCGTCGTCGTTG) and the W-box mutant mBS65 (5′-ATCGTCCTACGAGTCCTATTT-3′) (*Chujo et al., 2007*), all of which were labeled by Biotin. EMSA was performed using a LightShift Chemiluminescent EMSA Kit (Thermo, Rockford, IL) according to the manufacturer's instructions.

Competition experiments were performed using unlabeled BS65 as a competitor in a 250-fold molar excess.

## In vitro phosphorylation assay

GST-MPK3 (1 µg) or GST-MPK6 (1 µg) with or without His-MKK4$^{DD}$ (1 µg) was incubated in a kinase reaction buffer (50 mM Tris–HCl, pH 7.5, 1 mM Dithiothreitol (DTT), 10 mM MgCl$_2$, 10 mM MnCl$_2$, 50 µM ATP) at 30°C for 30 min. After this, recombinant His-WRKY70 (1 µg) was added and the mixture was incubated again at 30°C for 30 min. The reactions were stopped by adding SDS-loading buffer and heated for at 95°C for 5 min. The products were analyzed by SDS-PAGE. Phosphorylated proteins were detected by using Phos-tag Biotin BTL-104 (Wako, Japan) according to the manufacturer's instruction.

## QRT-PCR analysis

Five independent biological samples were used. Total RNA was isolated using the SV Total RNA Isolation System (Promega, Madison, WI). One µg of each total RNA sample was reverse transcribed using the PrimeScript RT-PCR Kit (TaKaRa). qRT-PCR was performed on a CFX96 Real-Time system (Bio-RAD, Richmond, CA) using a Premix Ex Taq Kit (TaKaRa). The primers and probe sequences used for mRNA detection of target genes by qRT-PCR are shown in *Supplementary file 1B*. A rice actin gene, *OsActin* (TIGR ID: Os03g50885), was used as an internal standard to normalize cDNA concentrations.

## SA, JA, and ET analysis

Plants (one per pot) were randomly assigned to SSB and control treatments. Two irWRKY70 lines (irWRKY70-7 and irWRKY70-8), two hemi-oeWRKY70 lines (hemi-oeWRKY70-8 and hemi-oeWRKY70-17), and one WT line were used. The stems were harvested at 0, 1.5, and 3 hr after SSB treatment, and JA and SA levels were analyzed by GC–MS using labeled internal standards as described previously (*Lu et al., 2011*). Three plants were covered with a sealed glass cylinder (diameter 4 cm, height 50 cm) and ethylene production was determined at 12 and 24 hr after the start of the experiment using the method described previously (*Lu et al., 2006*). Each treatment at each time interval was replicated 10 times.

## Quantification of endogenous GAs

10-day-old seedlings (3 g) of hemi-oeWRKY70-8, hemi-oeWRKY70-17, and WT were frozen in liquid nitrogen, ground to fine powder, and extracted with 15 ml of 80% (vol/vol) methanol at 48°C for 12 hr. Different [$_2$H$^2$] labeled GAs were added to plant samples before grinding as internal standards. The extraction and analysis were performed as described previously (*Chen et al., 2011*). Each line was replicated 3 times.

## TrypPI analysis

Plant stems (0.2–0.3 g per sample) were harvested at different time after different treatment (see details in *Figure 7*). The TrypPI activity was measured using a radial diffusion assay as described previously (*Van Dam et al., 2001*). Each treatment was replicated 5 times.

## Herbivore resistance experiments

### SSB performance assay

Three freshly hatched SSB larvae were allowed to feed on transgenic (irWRKY70 and hemi-OeWRKY70 lines) and WT plants. In complementation experiments, irWRKY70 lines (irWRKY70-7 and irWRKY70-8) and WT plants were randomly assigned to JA and Buffer treatments and then the freshly hatched SSB larvae were placed on these plants 24 hr after treatment. 30 plants were used for each line or treatment. Larval mass (to an accuracy of 0.1 mg) was measured 14 days after the start of the experiment. To detect the differences in plant tolerance to SSB attack between transgenic lines and WT plants, one second-instar larva of SSB was placed on individual plant. The damage levels of each plant were recorded and photographed every day.

### BPH performance assay

To investigate the colonization and oviposition behavior of BPH, the basal stem of two plants (a mutant plant vs a WT plant) in one pot were confined with glass cylinders into which 15 gravid BPH

females were introduced. The number of BPH on each plant at different time points and BPH eggs 48 hr post infestation were counted on each plant. To detect the survival rates of BPH nymphs on each line, the basal stem of each plant was confined with a glass cylinder, into which 15 BPH neonates were released. The number of surviving BPH on each plant was recorded until 12 days after the release of the herbivores. In $GA_3$ complementation experiments, hemi-oeWRKY70 lines (hemi-oeWRKY70-8 and/or hemi-oeWRKY70-17) and WT plants were used and these plants were randomly assigned to $GA_3$ and corresponding control treatments; the colonization and oviposition preferences of BPH female adults for pairs of plants and the survival rate of BPH nymphs on some treatments were determined (see details in *Figure 9*). The experiments for each treatment were replicated 8 times.

## Data analysis

Differences in plant height, root length, herbivore performance, expression levels of genes and JA, SA, GA, ethylene, and $H_2O_2$ levels on different treatments, lines, or treatment times were determined by analysis of variance (ANOVA) (Student's *t*-tests for comparing two treatments). All tests were carried out with Statistica (Statistica, SAS Institute Inc., http://www.sas.com/).

## Accession numbers

Sequence data from this article can be found in the Rice Annotation Project Database (RAP-DB) under the following accession numbers: Os05g39720 (*OsWRKY70*), Os05g27730 (*OsWRKY53*), Os01g61080 (*OsWRKY24*), Os03g17700 (*OsMPK3*), Os06g06090 (*OsMPK6*), Os02g54600 (*OsMKK4*), Os08g44590 (*OsGA20OX7*), Os08g39840 (*OsHI-LOX*), Os03g12500 (*OsAOS2*), Os09g19734 (*OsICS1*), Os04g48850 (*OsACS2*).

## Acknowledgements

We thank Tongfang Zhang and Ziyi Tang for their invaluable assistance with the experiments. The study was jointly sponsored by the National Basic Research Program of China (2010CB126200), the National Natural Science Foundation of China (31330065), and the earmarked fund for China Agriculture Research System (CARS-01-21). The work of ME is supported by the Swiss National Science Foundation (Grant Nr. 155781) and the European Commission (FP7-PEOPLE-2013-CIG- 629134).

## Additional information

### Funding

| Funder | Grant reference | Author |
|---|---|---|
| Ministry of Science and Technology of the People's Republic of China | National Basic Research Program of China (2010CB126200) | Yonggen Lou |
| National Natural Science Foundation of China | 31330065 | Yonggen Lou |
| Earmarked Fund for China Agriculture Research System | CARS-01-21 | Yonggen Lou |
| European Commission (EC) | FP7-PEOPLE-2013-CIG- 629134 | Matthias Erb |
| Schweizerische Nationalfonds zur Förderung der Wissenschaftlichen Forschung | 155781 | Matthias Erb |

The funders had no role in study design, data collection and interpretation, or the decision to submit the work for publication.

### Author contributions

RL, Conception and design, Acquisition of data, Analysis and interpretation of data, Drafting or revising the article, Contributed unpublished essential data or reagents; JZ, Acquisition of data, Analysis and interpretation of data, Drafting or revising the article, Contributed unpublished essential data or reagents; JL, Acquisition of data, Drafting or revising the article, Contributed unpublished essential data or reagents; GZ, QW, WB, Acquisition of data, Analysis and interpretation

of data, Drafting or revising the article; ME, Conception and design, Analysis and interpretation of data, Drafting or revising the article; YL, Conception and design, Analysis and interpretation of data, Drafting or revising the article, Contributed unpublished essential data or reagents

**Author ORCIDs**
Yonggen Lou, http://orcid.org/0000-0002-3262-6134

## Additional files

**Supplementary file**
• Supplementary file 1. (**A**) Primers used for cloning of full-length or partial cDNAs of target genes in this study. (**B**) Primers and probes used for QRT-PCR of target genes.

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
