## [Decision Letter]

[Editors’ note: this article was originally rejected after discussions between the reviewers, but the authors were invited to resubmit after an appeal against the decision.]

Thank you for choosing to send your work entitled “Prioritizing plant defence over growth facilitates infestation by non-target herbivores” for consideration at *eLife*. Your full submission has been evaluated by Deputy Editor Detlef Weigel, a member of our Board of Reviewing Editors, and two peer reviewers. The decision was reached after discussions between the reviewers. Based on our discussions and the individual reviews below, we regret to inform you that your work will not be considered further for publication in *eLife*.

The agreement was that the work is interesting, but does not represent a sufficiently major advance, as the molecular mechanisms underlying some of your key observations remain unclear. For example, it would have been important to demonstrate what the consequences of the observed physical interaction of OsWRKY70 with MAPK3/6 are. One would expect that, similar to what has been found for the *Arabidopsis* orthologue AtWRKY33, this interaction results in enhanced phosphorylation. But if this were the case, what would be the consequence of this phosphorlyation event: enhanced binding of OsWRKY33 to the W-box, degradation of OsWRKY70, or some other effect.

Reviewer #1:

This manuscript focuses on the function of the rice gene OsWRK70 in regulating plant defense and growth. OsWRK70 is a homolog of the *Arabidopsis* AtWRKY33 and *Nicotiana attenuata* NaWRKY3 and NaWRKY6 genes, which have previously been shown to function early in plant defense. The methodology seems correct and the conclusions do not go beyond what is shown by the data.

Specific comments:

1) In the current study, the authors present a number of findings: (1) OsWRK70 expression is increased by SSB feeding, but not jasmonate, salicylate, or ethylene; (2) OsWRK70 is localized to the nucleus, where it activates gene expression by binding to W-box motifs and physically interacts with OsMPK3 and OsMPK6, which are also required for up-regulating OsWRK70 transcription; (3) several approaches show that OsWRK70 positively regulates plant defense via JA and negatively regulates plant growth via GA; (4) high levels of OsWRK70 limit SSB growth but improve BPH growth. These findings are consistent with functions that have been reported previously for the corresponding *Arabidopsis thaliana* (AtWRKY33) and *Nicotiana attenuata* (NaWRKY3 and NaWRKY6) genes, as well as with earlier publications showing negative interactions between jasmonate and gibberellin signaling. Prior to the current manuscript, no publication has put these findings together in such a specific manner to illustrate a tradeoff in plant defense and growth based on a single transcription factor. Nevertheless, the manuscript as a whole seems more incremental than ground-breaking.

2) The section on regulation of OsWRK70 by OsMPK3 and OsMPK6 has some missing links. Although binding to OsWRK70 is demonstrated, the authors do not show that OsWRK70 is phosphorylated. I had expected some sort of demonstration that OsMPK3 and/or OsMPK6 modify OsWRK70 activity. However, the main phenotype that is demonstrated is reduced OsWRK70 transcription in lines with expression-silenced OsMPK3 and OsMPK6. Does this mean that OsWRK70 up-regulates its own transcription after binding to OsMPK3 and OsMPK6? Does the OsWRK70 have a W-box motif? Or do the authors have some other explanation for the transcriptional up-regulation.

3) Based on the authors' results, there seems to be a positive interaction between GA-mediated BPH defense and growth, i.e. stronger defense equals more growth and vice versa. This seems to run counter to the overall hypothesis of a trade-off between plant defense and growth. I realize that this is a bit peripheral to the main theme of the manuscript, but perhaps the authors can comment on this in the Discussion. Why is the rice BPH interaction so different from the rice-SSB interaction?

Reviewer #2:

The paper reports on the role of a rice transcription factor, OsWRKY70, in modulating resistance/susceptibility towards the chewing herbivore *C. suppressalis* (rice striped stem borer SSB) and the rice brown plant hopper *N. lugens* (BPH). The presented data imply that OsWRKY70, which is induced upon SSB challenge but not by BPH, positively regulates JA/ET levels, and functions upstream of JA and ET signaling. Using two independent RNAi knock-down and OsWRKY70 overexpressor (OE) lines each, the authors show that OsWRKY70 positively regulates resistance to SSB but negatively affects resistance to BPH. The authors demonstrate that OsWRKY70 physically interacts with OsMPK3 and OsMPK6, and that silencing of these MAP kinases results in decreased levels of OsWRKY70 transcript. OsWRKY70-mediated SSB resistance very likely occurs via increased JA-dependent trypsin protease inhibitor activities. OsWRKY70 also appears to be a negative regulator of GA biosynthesis since OsWRKY70 OE lines show significantly lower GA levels when compared to wildtype plants. The negative regulation of BPH resistance by OsWRKY70 appears to be through GA signaling.

Overall the paper contains interesting results and the experiments in most cases appear to be properly performed and documented. However there are several issues that need to be addressed. My major concern however, is that several key aspects of the paper rely mainly on data obtained by gross overexpression of the OsWRKY70 transcription factor. These plants (even the hemizygous lines used in the study) show strong pleiotropic affects. Thus, how valid some of uncovered molecular mechanisms are under normal physiological conditions remains an open question.

Specific points:

1) As a general comment I feel that the authors should replace the abbreviations often used in the different figures to make it easier for the reader to follow the story. For example: In Figure 2 and Figure 3 the silencing lines for MPK6 are designated as irMPK6-1 and irMPK6-2 but similar lines for MPK3 are designated P3i-53 and P3i-183. Suggestion: replace this by irMPK3-53 and irMPK3-183.

The OsWRKY70 silencing lines are designated ir7 and ir8, whereas the hemizygous OsWRKY70 overexpressor lines are designated hemi-oe8 and hemi-oe17 in various figures. I would suggest obvious names such as irWRLY70-7, irWRKY70-8, oeWRKY70-8, and oeWRKY70-17.

2) Figure 2. The pull-down data concerning interaction of OsWRKY70 with OsMPK6 is not convincing. Considering that the GST-MPK6 protein is larger than the GST-MPK3 protein on the Coomassie stained gel, I do not find the weak band detected by the anti-WRKY70 antibody for GST-MPK6 on the western blot to be convincing particularly as it appears to be of the same size as the GST-MPK3 band. This needs improvement.

3) Figure 2—figure supplement 1. The gel is of very poor quality and needs to be improved. Detection of the shifted band is very weak and is also still somewhat observed when the mutated probe (OsWRKY70 + mBS65) is used. Moreover, the EMSA experiments are not described in Materials and methods.

4) I do not feel that the transcriptional profiling data actually adds much to the paper. These experiments were performed using the OsWRKY70 overexpressor plants not infested with the herbivore. Thus, only a few genes were identified whose expression was solely dependent on OsWRKY70. It is not clear to me why SSB-induced WT plants, and OsWRKY70 overexpressor plants induced by SSB were not included. Indeed, nearly none of the genes further analyzed (Figure 4—figure supplement 1, Figure 5) in later parts of the paper *(OsHI-LOX, OsAOS2, OsACS2, OsWRKY24, OsWRKY53*) were detected (see Table 1 and 2). Are any of the identified differentially expressed genes depicted in Figure 4—figure supplement 2 altered in their expression upon SSB infestation?

5) Were GA levels in the RNAi lines tested? Are GA levels higher than WT in such plants?

6) Figure 5—figure supplement 1: Not all W-box elements highlighted are true functional elements. The W-box is not merely defined by the core TGAC sequence shown.

7) There is substantial variation for the WT values in the experiments depicted in Figure 8. For example in Figure 8 BPH survival rates on WT decline from 100% to 60% within 12 days in this set of experiments (that include the overexpressors), but only decline from 100% to 85% in the experiments shown in Figure 8 (that include the RNAi lines). Why is this? Are these types of experiments highly variable and if so, how confident can one be about such results?

---

## [Author Response]

Based on your comments, we decided to perform a series of additional experiments to elucidate the molecular mechanisms of WRK70 regulation in rice. Our new results show that two map kinases can phosphorylate WRKY70 and thereby enhance its transactivation activity independently of W‐box autoregulation.

These new experiments allow us to directly address the main criticism of the paper. We furthermore believe that all other comments by the reviewers could easily be addressed and resolved.

*The agreement was that the work is interesting, but does not represent a sufficiently major advance, as the molecular mechanisms underlying some of your key observations remain unclear. For example, it would have been important to demonstrate what the consequences of the observed physical interaction of OsWRKY70 with MAPK3/6 are. One would expect that, similar to what has been found for the* Arabidopsis *orthologue AtWRKY33, this interaction results in enhanced phosphorylation. But if this were the case, what would be the consequence of this phosphorlyation event: enhanced binding of OsWRKY33 to the W-box, degradation of OsWRKY70, or some other effect*.

We agree with the editors that demonstrating the consequences of OsWRKY70 and MAPK3/6 binding would be interesting. We have now done these experiments and found that MPK3/6 can phosphorylate OsWRKY70 and thereby enhance the transactivation activity of OsWRKY70, but not the binding activity of OsWRKY70 to its own W‐box. We have added these results in Figure 2.

Reviewer #1:

*This manuscript focuses on the function of the rice gene OsWRK70 in regulating plant defense and growth. OsWRK70 is a homolog of the* Arabidopsis *AtWRKY33 and* Nicotiana attenuata *NaWRKY3 and NaWRKY6 genes, which have previously been shown to function early in plant defense. The methodology seems correct and the conclusions do not go beyond what is shown by the data.*

*Specific comments*:

*1) In the current study, the authors present a number of findings: (1) OsWRK70 expression is increased by SSB feeding, but not jasmonate, salicylate, or ethylene; (2) OsWRK70 is localized to the nucleus, where it activates gene expression by binding to W-box motifs and physically interacts with OsMPK3 and OsMPK6, which are also required for up-regulating OsWRK70 transcription; (3) several approaches show that OsWRK70 positively regulates plant defense via JA and negatively regulates plant growth via GA; (4) high levels of OsWRK70 limit SSB growth but improve BPH growth. These findings are consistent with functions that have been reported previously for the corresponding* Arabidopsis thaliana *(AtWRKY33) and* Nicotiana attenuata *(NaWRKY3 and NaWRKY6) genes, as well as with earlier publications showing negative interactions between jasmonate and gibberellin signaling. Prior to the current manuscript, no publication has put these findings together in such a specific manner to illustrate a tradeoff in plant defense and growth based on a single transcription factor. Nevertheless, the manuscript as a whole seems more incremental than ground-breaking*.

We agree with the reviewer that there is overlap between the function of OsWRKY70, NaWRKY3 and NaWRKY6 of *Nicotiana attenuata* and AtWRKY33 of *Arabidopsis thaliana*. Also, negative interactions between jasmonate and gibberellin signaling via COI1-JAZ1-DELLA-PIF complexes have been documented. The strength of our paper is the combination of mechanistic depth with biological relevance. On a mechanistic level, we show for the first time that a WRKY transcription factor directly regulates GA-JA crosstalk. On a biological level, we demonstrate that this regulation is responsible for growth defense trade-offs. Furthermore, we demonstrate that prioritizing defense over growth against one herbivore also leads to an additional cost by reducing resistance against another herbivore. In our opinion, this is not an incremental finding, but a new mechanism for defense trade-offs in plants which has important implications for the evolution and agricultural exploitation of induced resistance.

*2) The section on regulation of OsWRK70 by OsMPK3 and OsMPK6 has some missing links. Although binding to OsWRK70 is demonstrated, the authors do not show that OsWRK70 is phosphorylated. I had expected some sort of demonstration that OsMPK3 and/or OsMPK6 modify OsWRK70 activity. However, the main phenotype that is demonstrated is reduced OsWRK70 transcription in lines with expression-silenced OsMPK3 and OsMPK6. Does this mean that OsWRK70 up-regulates its own transcription after binding to OsMPK3 and OsMPK6? Does the OsWRK70 have a W-box motif? Or do the authors have some other explanation for the transcriptional up-regulation*.

We appreciate this comment. It has been well documented that Group I-type WRKY TFs can be phosphorylated by MAPKs and that the SP clusters are the phosphorylating sites (Ishihama et al., 2012; [27]). Like *N. benthamiana* NbWRKY8, *Arabiodpsis* AtWRKY33 and rice OsWRKY53, OsWRKY70 also has SP clusters (Figure 1—figure supplement 1). Here, we observed that OsWRKY70 is phosphorylated by OsMAPK3 and OsMAPK6 via their physical interactions, and this phosphorylation enhances the thansactivation activity of OsWRKY70 (Figure 2). Moreover, OsWRKY70 contains W-boxes in its promoter region and OsWRKY70 can auto-regulate its expression by binding its own promoter (Figure 2—figure supplement 2). Therefore, OsWRKY70 transcript levels are likely reduced in OsMAPK3 and 6 silenced lines because of the reduction in phosphorylated WRKY70 and other WRKYs, which decreases WRKY activity and thereby reduces OsWRKY70 transcript levels.

*3) Based on the authors' results, there seems to be a positive interaction between GA-mediated BPH defense and growth, i.e. stronger defense equals more growth and vice versa. This seems to run counter to the overall hypothesis of a trade-off between plant defense and growth. I realize that this is a bit peripheral to the main theme of the manuscript, but perhaps the authors can comment on this in the Discussion*. *Why is the rice BPH interaction so different from the rice-SSB interaction?*

We thank the reviewer for this valuable idea. This aspect may be related to the different feeding modes of the two herbivores. We now discuss this aspect at the end of the Discussion section.

Reviewer #2:

*[…] Overall the paper contains interesting results and the experiments in most cases appear to be properly performed and documented. However there are several issues that need to be addressed. My major concern however, is that several key aspects of the paper rely mainly on data obtained by gross overexpression of the OsWRKY70 transcription factor. These plants (even the hemizygous lines used in the study) show strong pleiotropic affects. Thus, how valid some of uncovered molecular mechanisms are under normal physiological conditions remains an open question*.

We agree with the reviewer that growth-defense trade-offs can be challenging to investigate because of potential secondary effects of the growth phenotypes, especially when it comes to herbivore resistance effects. To overcome this problem, we used three different strategies. First, we used hemizygous OsWRKY70 overexpressing lines, which show a much weaker growth phenotype. Second, we created both OsWRKY70 overexpressing and silenced lines. The silenced lines did not show any growth phenotype and therefore do not suffer from any growth-related limitations. Third, we have performed hormonal complementation assays to demonstrate the causal relation between OsWRKY70 dependent signaling and herbivore resistance. Together, we are confident that the reported mechanisms and effects of OsWRKY are robust and valid.

*Specific points*:

*1) As a general comment I feel that the authors should replace the abbreviations often used in the different figures to make it easier for the reader to follow the story. For example: In*
Figure 2
*and*
Figure 3
*the silencing lines for MPK6 are designated as irMPK6-1 and irMPK6-2 but similar lines for MPK3 are designated P3i-53 and P3i-183. Suggestion: replace this by irMPK3-53 and irMPK3-183. The OsWRKY70 silencing lines are designated ir7 and ir8, whereas the hemizygous OsWRKY70 overexpressor lines are designated hemi-oe8 and hemi-oe17 in various figures. I would suggest obvious names such as irWRLY70-7, irWRKY70-8, oeWRKY70-8, and oeWRKY70-17*.

We thank the reviewer for this valuable suggestion. We have changed the names of the mutants as the reviewer suggested.

*2)*
Figure 2*. The pull-down data concerning interaction of OsWRKY70 with OsMPK6 is not convincing. Considering that the GST-MPK6 protein is larger than the GST-MPK3 protein on the Coomassie stained gel, I do not find the weak band detected by the anti-WRKY70 antibody for GST-MPK6 on the western blot to be convincing particularly as it appears to be of the same size as the GST-MPK3 band. This needs improvement*.

We used GST proteins (GST-MPK3 or GST-MPK6) to pull down the HIS-OsWRKY70 protein; after that, the protein complexes were boiled in SDS-loading buffer and then were fractionated by SDS-PAGE. The anti-WRKY70 antibody was used to investigate if there is the HIS-WRKY70 protein in the protein complexes. If there is, this means that the GST proteins can bind to the HIS-WRKY70. Thus, the western blot bands showed in Figure 2 are all the HIS-OsWRKY70 protein; they should have the same size. We therefore think that the conclusions of this experiment are valid.

*3)*
Figure 2—figure supplement 1*. The gel is of very poor quality and needs to be improved. Detection of the shifted band is very weak and is also still somewhat observed when the mutated probe (OsWRKY70 + mBS65) is used. Moreover, the EMSA experiments are not described in Materials and methods*.

We thank the reviewer for this comment. We have redone the experiment and replaced the gel picture. We have added the method of EMSA in the Materials and methods section.

*4) I do not feel that the transcriptional profiling data actually adds much to the paper. These experiments were performed using the OsWRKY70 overexpressor plants not infested with the herbivore. Thus, only a few genes were identified whose expression was solely dependent on OsWRKY70. It is not clear to me why SSB-induced WT plants, and OsWRKY70 overexpressor plants induced by SSB were not included. Indeed, nearly none of the genes further analyzed (*Figure 4—figure supplement 1*,*
Figure 5*) in later parts of the paper (*OsHI-LOX, OsAOS2, OsACS2, OsWRKY24, OsWRKY53*) were detected (see Table 1 and 2). Are any of the identified differentially expressed genes depicted in*
Figure 4—figure supplement 2
*altered in their expression upon SSB infestation?*

We thank the reviewer for this question. Since overexpression of OsWRKY70 resulted in a strong growth phenotype, we first wanted to know what happened in plants when OsWRKY70 was constitutively overexpressed- We then used the main patterns of this profiling to investigate specific effects of OsWRKY70 in more detail, including, for instance, GA signaling. Thus, the data is important to understand the motivation for many of the follow-up experiments. It would indeed be interesting in the SSB-induced regulation of some of the potential OsWRKY70 regulated genes. However, this experiment is beyond the scope of the current study.

5) Were GA levels in the RNAi lines tested? Are GA levels higher than WT in such plants?

We did not check the GA levels in RNAi lines, We did not check the GA levels in RNAi lines. Since GAs are correlated to plant growth and the RNAi lines showed similar growth phenotypes to WT plants, it is unlikely that GA levels were altered in these lines. This suggests an involvement of other factors, such as the homologs of OsWRKY70, OsWRKY24 and OsWRKY53, in the biosynthesis of GAs. This aspect is discussed in the revised version of the paper.

*6)*
Figure 5—figure supplement 1*: Not all W-box elements highlighted are true functional elements. The W-box is not merely defined by the core TGAC sequence shown*.

We thank the reviewer for this valuable suggestion. We have improved the description in the figure legends.

*7) There is substantial variation for the WT values in the experiments depicted in*
Figure 8*. For example in*
Figure 8
*BPH survival rates on WT decline from 100% to 60% within 12 days in this set of experiments (that include the overexpressors), but only decline from 100% to 85% in the experiments shown in*
Figure 8
*(that include the RNAi lines)*. *Why is this? Are these types of experiments highly variable and if so, how confident can one be about such results?*